



# Statistical characteristics of raindrop size distribution during rainy seasons in Beijing urban area and implications for radar rainfall estimation

Yu Ma[1], Guangheng Ni[1], V. Chandrasekar[2], Fuqiang Tian[1], Haonan Chen[2,3]

[1]State Key Laboratory of Hydro-Science and Engineering, Department of Hydraulic Engineering, Tsinghua University, Beijing 100084, China
[2]Colorado State University, Fort Collins, CO 80523, USA
[3] NOAA/Earth System Research Laboratory, Boulder, CO 80305, USA

*Correspondence to*: Haonan Chen (haonan.chen@noaa.gov)

**Abstract.** Raindrop size distribution (DSD) information is fundamental in understanding the precipitation microphysics and quantitative precipitation estimation, especially in complex terrain or urban environment which is known for its complicated rainfall mechanism and high spatial and temporal variability. In this study, the DSD characteristics of rainy seasons in Beijing urban area are extensively investigated using 5-year DSD observations from a Parsivel[2] disdrometer located at Tsinghua University. The results show that the DSD samples with rain rate < 1 mm h$^{-1}$ account for more than half of total observations. The mean values of $\log_{10}N_w$ and $D_m$ of convective rain are higher than that of stratiform rain, and there is a clear boundary between the two types of rain in terms of the scattergram of $\log_{10}N_w$ versus $D_m$. The convective rain in Beijing is neither continental nor maritime owing to the particular location and local topography. As the rainfall intensity increases, the DSD spectra become higher and wider, but they still have peaks around diameter $D \sim 0.5$ mm. The midsize drops contribute most towards accumulated rainwater. The $D_m$ and $\log_{10}N_w$ values show a diurnal cycle and an annual cycle. In addition, DSD shows higher $D_m$ values and lower $\log_{10}N_w$ values during the periods of strong urban heat island (UHI) effect and UHI up stage of a day, and the same in July and August. The localized radar reflectivity ($Z$) and rain rate ($R$) relations ($Z = aR^b$) show substantial differences compared to the commonly used NEXRAD relationships. And the polarimetric radar algorithms $R(K_{dp})$, $R(K_{dp}, Z_{DR})$, and $R(Z_H, Z_{DR})$ show greater potential for rainfall estimation.

## 1 Introduction

Raindrop size distribution (DSD) provides fundamental information on precipitation microphysics. Understanding the DSD variability is of great importance in remote sensing observations of precipitation and microphysical parameterizations in numerical weather prediction (NWP) models. For example, the DSD serves as a fundamental bridge in deriving the Z-R relationships used by ground based weather radar (Battan, 1973; Uijlenhoet and Stricker, 1999) and space borne radar (i.e.

TRMM PR: Iguchi et al., 2000; and GPM DPR: Hou et al., 2014) for quantitative precipitation estimation (QPE). The NWP





systems coupled with various DSD models can capture more detailed horizontal and/or vertical rain information so as to improve the accuracy of precipitation predictions (Abel and Boutle, 2012; Fadnavis et al., 2014; McFarquhar et al., 2015; Saleeby and Cotton, 2004). In addition, the DSD is highly related to the kinetic energy of rainfall that has substantial impact on the soil erosions (Angulo-Martinez and Barros, 2015; Caracciolo et al., 2011; Ellison, 1945; Kinnell, 2005; Lim et al.,

2015), which is very useful in further understanding of runoff processes and mitigation of subsequent flood hazards (Angulo-Martinez and Barros, 2015; Smith et al., 2009).

Numerous studies have been devoted to the statistical characteristics of DSD worldwide. It is found that the DSD characteristics vary with geographical locations, climate regimes, seasons, rain types, and even diurnal cycles (Dolan et al., 2018; Seela et al., 2018; Tokay and Short, 1996; Wen et al., 2017b). Dolan et al. (2018) classified the global DSD characteristics into six groups

by analyzing 12 global disdrometer datasets across three latitudes using principal component analysis. They found that the physical processes shaping the DSD characteristics were likely to vary as a function of location. The comparison of DSD in northern and southern China in Tang et al. (2014) showed that there was a clear difference in precipitation microphysical parameters between different regimes during convective rain, while the difference was less notable for stratiform events. The DSD analysis in Beijing (Wen et al., 2017a) and Taiwan (Seela et al., 2018) also indicated that there were significant

differences in DSD between summer and winter rainfall, and both showed the diurnal variation. In addition, the DSD may exhibit high variability in special weather systems. For example, DSD of the tropical cyclones has a higher concentration of small and middle size drops as well as a lower mass-weighted mean diameter (i.e., $D_m$) in all types of rain compared with the non-tropical cyclone in Darwin (Deo and Walsh, 2016).

Beijing, the capital of China, is a very densely populated metroplex with a population higher than 21 million. It is more

vulnerable to extreme weather events such as extreme rainfall and floods (Zhang et al., 2013). Rainfall monitoring networks with high-temporal and spatial resolution (e.g., dense automatic weather stations network de Vos et al., 2017; remote sensing network described by Chen and Chandrasekar 2015 and Cifelli et al., 2018) have been applied in several metropolitan areas, as the hydrology response in urban area is sensitive to the spatial and temporal variability of rainfall (Cristiano et al., 2017). The precipitation in Beijing is more complex with high spatial and temporal variability due to the combined effects of high-

urbanization and local unique topography (Song et al., 2014; Yang et al., 2013a; Yang et al., 2016), which highlights the importance of further understanding of DSD characteristics for enhanced urban precipitation measurements and modelling. Several studies on DSD characteristics in Beijing area have been conducted. Tang et al. (2014) studied the DSD characteristics and the polarimetric radar parameters for convective and stratiform rain from July to October 2008 in Beijing and compared with other regions using a first-generation laser-based optical particle size and velocity (Parsivel[1]) disdrometer produced by

OTT Messtechnik, Germany. Wen et al. (2017a) investigated the statistical properties of summer and winter precipitation in Beijing, including the bulk properties, raindrop fall velocity, axis ratio, and DSD, using a two-dimensional video disdrometer (2DVD) and a micro-rain radar (MRR). However, these studies are mainly focused on summer time (June-September or July-October) and with very limited measurements from one season or two, which are not sufficient to represent DSD characteristics of Beijing during the rainy or warn seasons ranging from May to October. In addition, the DSD measurements used in previous





studies are more likely collected in the suburban area, which could not show the connections of DSD and the urbanization of Beijing.

This paper presents a comprehensive study of DSD properties using 5-year (2014–2018) continuous observations in Beijing urban area, aiming to advance our understanding of local rainfall microphysics and parameterization in remote sensing

retrievals and NWP models. This paper is organized as follows. Section 2 describes the dataset and methodologies for data quality control and analysis. The characteristics of DSD parameters for all rainfall events combined, different rainfall types, different classifications, different periods of a day, and different months are detailed in section 3. Section 4 presents the implications for radar QPE and the parameterization errors of different DSD-based radar rainfall algorithms. Summary and conclusion are given in section 5.

## 2 Data and Method

### 2.1 Dataset

In this study, a second-generation Particle Size and Velocity (Parsivel[2]) disdrometer is used, which is deployed at Tsinghua University campus, Beijing, China (hereafter referred to as THUD). Figure 1 illustrates the specific location of THUD (40.002 °N, 116.324 °E; 91m above sea level), relative to the Beijing metroplex. It is an optical disdrometer with a 54 cm$^2$

horizontal sample area and configured with 1 minute sampling resolution to measure the DSD and fall velocity of rain drops (Löffler-Mang and Joss, 2000). The velocity and particle size are divided into 32 non-uniform bins, varying from 0.05 to 20.8 m s$^{-1}$ for velocity and 0.062 to 24.5 mm for drop diameter.

The DSD measurements are collected from June 2014 to December 2018. Lyu et al. (2018) compared the accumulated rainfall computed from DSD data with the rainfall measured by an automatic weather station 350 m away from THUD, to cross-check

the reliability of both instruments. Since most rainfall in Beijing area occurs during rainy season which usually begins from May to the end of October (Song et al., 2014), this study uses the data collected from May to October to analyze the DSD characteristics.

### 2.2 Method

The direct measurements from disdrometer are the number of raindrops at each velocity ($i$) and diameter ($j$) bin. From the data,

the maximum diameter $D_{max}$ (mm) of raindrops can be obtained directly, and the total number of rain drops $T_d$ can be calculated:

$$T_d = \sum_{i=1}^{32} \sum_{j=1}^{32} n_{i,j}, \tag{1}$$

where $n_{i,j}$ stands for the drop number at each bin.

The number concentration of raindrops per unit volume for the $j$th diameter bin can be calculated as follows:

$$N(D_j) = \sum_{i=1}^{32} \frac{n_{i,j}}{A \cdot \Delta t \cdot V_i \cdot \Delta D_j}, \tag{2}$$





where $N(D_j)$ is in $m^{-3}$ $mm^{-1}$; $A$ is the sampling area in $m^2$; $\Delta t$ is the sampling time interval in s; $A$ and $\Delta t$ are respectively 0.0054 $m^2$ and 60 s in this study; $\Delta D_j$ (mm) is the diameter interval from $D_j$ to $D_{j+1}$ for the $j$th diameter bin; $V_i$ (m $s^{-1}$) is the fall speed for the $i$th velocity class. Because of the measurement error, especially for larger size drops (Tokay et al., 2014), the empirical terminal velocity–diameter ($V - D$) relationship in Atlas et al. (1973) is adopted in this study:

$$V(D_j) = 9.65 - 10.3\exp(-0.6D_j), \tag{3}$$

The Gamma form DSD (Ulbrich, 1983) in the following form has been proved to be suitable to describe the raindrop spectra.

$$N(D) = N_0 D^\mu \exp(-\Lambda D), \tag{4}$$

where $D$ (mm) is the rain drop diameter; $N(D)$ ($mm^{-1}$ $m^{-3}$) is the number concentration of raindrops per unit volume and per diameter interval; $N_0$ ($mm^{-1-\mu}$ $m^{-3}$), $\mu$ and $\Lambda$ are the scale, shape and slope parameters of Gamma distribution, and these three parameters can be derived using gamma moments (GM) (Kozu and Nakamura, 1991; Tokay and Short, 1996) or maximum likelihood methods (Montopoli et al., 2008). When $\mu=0$, the Gamma form DSD degenerates into an exponential DSD model. In this study, we use the normalized gamma DSD described by Testud et al. (2000) which is commonly used to describe the natural variations of DSD (e.g.: Bringi and Chandrasekar 2001; Dolan et al., 2018).

$$N(D) = N_w f(\mu) \left(\frac{D}{D_m}\right)^\mu \exp\left[-(4+\mu)\frac{D}{D_m}\right], \tag{5}$$

where $N_w$ ($m^{-3}$ $mm^{-1}$) is normalized intercept parameter; $D_m$ (mm) is the mass-weighted mean diameter. $N_w$, $D_m$, and $f(\mu)$ are calculated as follows:

$$D_m = \frac{\sum_{j=1}^{32} N(D_j) \cdot D_j^4 \cdot \Delta D_j}{\sum_{j=1}^{32} N(D_j) \cdot D_j^3 \cdot \Delta D_j}, \tag{6}$$

$$N_w = \frac{4^4}{\pi \rho_w} \left(\frac{10^3 W}{D_m^4}\right), \tag{7}$$

$$f(\mu) = \frac{6(4+\mu)^{\mu+4}}{4^4 \Gamma(\mu+4)}, \tag{8}$$

The integral parameters of total number concentration $N_t$ ($m^{-3}$), rain rate $R$ (mm $h^{-1}$), liquid water content $W$ (g $m^{-3}$) and the mass spectrum standard deviation $\sigma_m$ (mm) are also calculated in this study based on the following equations.

$$N_t = \sum_{i=i}^{32} \sum_{j=1}^{32} \frac{n_{i,j}}{A \cdot \Delta t \cdot V_i}, \tag{9}$$

$$R = \frac{6\pi}{10^4 \rho_w} \sum_{j=1}^{32} V(D_j) D_j^3 N(D_j) \Delta D_j = \sum_{j=1}^{32} R(D_j) \Delta D_j, \tag{10}$$

$$W = \frac{\pi \rho_w}{6 \times 10^3} \sum_{j=1}^{32} D_j^3 N(D_j) \Delta D_j, \tag{11}$$

$$\sigma_m = \sqrt{\frac{\sum_{j=1}^{32} (D_j - D_m)^2 N(D_j) \cdot D_j^3 \cdot \Delta D_j}{\sum_{j=1}^{32} N(D_j) \cdot D_j^3 \cdot \Delta D_j}}, \tag{12}$$

where $\rho_w$ is the water density (1.0 g $cm^{-3}$); $R(D_j)$ (mm $h^{-1}$ $mm^{-1}$) is the rain rate at the $j$th diameter class, and it is normalized by the total rain rate $R$ as $R(D_j)^{norm} = \frac{R(D_j)}{R}$ in the analysis to make the comparison at different rain intensities more meaningful.





The median volume diameter $D_0$ (mm), defined such that drops smaller than $D_0$ contribute to half the total liquid water content ($W$), as follows:

$$\frac{\pi \rho_w}{6 \times 10^3} \int_0^{D_0} D^3 N(D) dD = \frac{1}{2} \frac{\pi \rho_w}{6 \times 10^3} \int_0^{\infty} D^3 N(D) dD = \frac{1}{2}(W),$$ (13)

is also computed and included in the analysis.

Considering that a high-resolution dual-polarization X-band radar network is being deployed in Beijing for urban hydrometeorological applications, a series of polarimetric radar variables are simulated at X-band frequency based on the DSD measurements using the T-matrix method (Waterman, 1965), including horizontal reflectivity $Z_H$ (mm⁶ m⁻³), differential reflectivity $Z_{dr}$ (dB), and specific differential phase $K_{dp}$ (°km⁻¹). The drop shape model used in the simulation is the one proposed by Thurai et al. (2007). The temperature data are obtained from an automatic weather station collocated with THUD

disdrometer. In addition, various DSD-based radar QPE relations are derived and their parameterization errors are investigated for future development of Beijing urban radar rainfall system.

**2.3 Quality Control**

To minimize the measurement errors and improve data reliability, several quality control procedures have been applied on the 1-minute DSD data. First, because of the low signal-to-noise ratios, the lowest diameter bins are not used which means the

raindrops less than 0.312 mm are eliminated in the analysis. Second, the 1-minute sample data with total raindrop number smaller than 10 or the derived rain rate less than 0.1 mm h⁻¹ are considered as noise and removed (Sreekanth et al., 2017). Then, if the continuous data satisfying the above conditions last less than 5 minutes, they will be ignored to avoid the spurious and erratic measurements (Jash et al., 2019). In addition, to focus on rainfall, all the data contaminated by hail are removed, and raindrops at a diameter of larger than 8 mm are eliminated since the biggest raindrops ever reported in the literature are

around 8 mm (Baumgardner and Colpitt, 1995; Beard et al., 1986). Also, thresholds on simulated radar parameters (i.e., $Z_h <$ 55 dBZ, $Z_{dr} > 0$ dB, and $K_{dp} > 0$ °km⁻¹) are used to further guarantee the creditability of the measured rainfall DSD data.

**3 DSD parameter characteristics**

**3.1 Distribution of DSD parameters**

A total number of 43618 1-minute DSD spectra have been selected after data quality control, covering the wet seasons (May

to October) from 2014 to 2018 except for May 2014 (no observation yet). In this study, the raindrops below 1 mm are considered small drops; 1–3 mm are midsize drops; and large drops if larger than 3 mm (Krishna et al., 2016; Seela et al., 2017; Seela et al., 2018; Tokay et al., 2014). The distribution and statistics of the DSD parameters are shown in Fig. 2 and Table 1. $D_0$ and $D_m$ have similar distribution with each other. Though $D_0$ has a larger range with a larger maximum and a smaller minimum value, it is more concentrated to small values, showing smaller mean and median diameter value with higher standard

deviation, skewness and kurtosis values. The relationship $\Lambda D_m + 3.67 = \Lambda D_0 + 4$ may explain for such phenomenon when





$\Lambda > 0$. The distribution of $D_{max}$ shows that in most of the rain events, the biggest drops are middle class size, indicating that most of the rainfall is potentially made up of small and moderate raindrops. The statistical characteristics of $\log_{10}N_w$ show almost equal median (3.596) and mean values (3.595), as well as a very small skewness value (0.040), indicating that $\log_{10}N_w$ follows a symmetry distribution. The mean, median and skewness values of $\log_{10}N_t$, $\log_{10}T_d$, and $\log_{10}\sigma_m$ also exhibit

symmetry distributions. Moreover, the kurtosis of these three parameters are close to 3, — characteristic of a normal distribution, which indicates that $N_t$, $T_d$, and $\sigma_m$ obey the lognormal distribution. Since a threshold of 0.1 mm h$^{-1}$ is applied on the rain rate field (i.e., $\log_{10}R$ is truncated by -1), the $R$ meets a positive skew distribution. Because of this, the distributions of $\log_{10}W$ also have a positive skew distribution. It is worth noting that DSD samples with rain rate about 0.8–1 mm h$^{-1}$ have the highest frequency and samples with rain rate less than 1 mm h$^{-1}$ account for more than half of total rain.

**3.2 DSD properties for different rain types**

There may be significant differences between the two general precipitation types (i.e., convective and stratiform), which results in variations in DSD characteristics. Previous studies in different climatic regions have shown that DSD differences in the two rain types have a great impact on parameterization in both NWP models and remote sensing observations. In this study, rainfall events are separated into stratiform and convective cases using a method combining Bringi et al. (2003) and Chen et al. (2013).

In particular, if the standard derivation of rain rate for a consequent 10-minute is greater than 1.5 mm h$^{-1}$ and the rain rate is greater than 5 mm h$^{-1}$, it is classified as convective rain, otherwise as stratiform rain.

Figure 3 shows the histograms of $D_m$ and $\log_{10}N_w$ for all the rainfall events and for the convective and stratiform subsets. The three key parameters including mean, standard deviation (SD), and skewness are also indicted in Fig. 3. For the total data set (Fig.3a), the $D_m$ histogram is highly positively skewed, while the skewness of $\log_{10}N_w$ is near to zero, suggesting that the

distribution of $\log_{10}N_w$ is more symmetrical. The standard deviations of $D_m$ and $\log_{10}N_w$ are large (0.46 mm for $D_m$ and 0.62 for $\log_{10}N_w$), indicating a high variability of both $D_m$ and $\log_{10}N_w$. The mean values of $D_m$ and $\log_{10}N_w$ are 1.15 mm and 3.60 respectively. It should be noted that both mean values are slightly smaller compared with those obtained in Beijing area during the summer time of 2015 (from 30$^{th}$ July to 30$^{th}$ September) and 2016 (from 9$^{th}$ June to 26$^{th}$ September) (Wen et al., 2017a), which means that the DSD during summer time may be more concentrated than the whole rainy seasons.

Considering different rain types, it can be found that the $D_m$ for both types are positively skewed, while the skewness of $\log_{10}N_w$ for convective exhibits negative. The spread of $\log_{10}N_w$ for convective rain is narrower compared to that of stratiform rain, and the skewness of $\log_{10}N_w$ is larger than that of stratiform rain ($-0.98$ versus 0.10). The spreads and skewness of $D_m$ for these two rainfall types perform oppositely (see Fig. 3 b–c). In addition, histograms of $D_m$ and $\log_{10}N_w$ during convective rain tend to shift toward the large values relative to stratiform rain, indicating that convective events have higher $D_m$ and

$\log_{10}N_w$ values than stratiform cases (1.91 mm and 3.66 for convective versus 1.08 mm and 3.59 for stratiform, respectively). As Fig. 4 shows, in both convective and stratiform rains, with the increase of rain rate, the $D_m$ increases (the positive exponents of the fitted power-law relationships), but the distributions of $D_m$ become narrowed. Note that at higher rain rate, the $D_m$





values tend to be a stable value, indicating that the DSD may have come to an equilibrium state where the coalescence and breakup of raindrops are in near balance (Hu and Srivastava, 1995). It can be seen in Fig. 4a, when rain rate $R > 60$ mm h$^{-1}$, the $D_m$ values reach a stable value around 2–2.5 mm, which means the increase of rain rate is mainly caused by an increase of concentration (Bringi and Chandrasekar, 2001). With respect to the $D_m - R$ relationship, the coefficient and exponent values

of convective rain are slightly higher than stratiform, suggesting a larger $D_m$ of convective rain than that of stratiform rain for a given rain rate, which draws an opposed conclusion to Wen et al. (2016) in eastern China and Zhang et al. (2019a) in southern China.

Figure 5 shows the distribution of $\log_{10} N_w$ versus $D_m$ derived from the DSD data for all the rainfall events, as well as two different rain types. The statistical results from different parts of China (i.e., North China, East China, and South China)

reported by Chen et al. (2013), Tang et al. (2014), Wen et al. (2016), Wen et al. (2017a), and Zhang et al. (2019a) are also indicated in Fig. 5. The two black rectangles correspond to the maritime and continental convective clusters and the dashed line corresponds to the stratiform (hereafter called "stratiform line") case described by Bringi et al. (2003). For all the events combined, the distribution has a wide scale, but most points concentrate in the area of high $\log_{10} N_w$ with low $D_m$. For convective and stratiform events, the distributions are concentrated in different areas (Stratiform: 3.3–4.0 for $\log_{10} N_w$, 0.8–

1.2 mm for $D_m$; Convective: 3.7–4.2 for $\log_{10} N_w$, 1.4–2.0 mm for $D_m$). There is a clear boundary between the two types, even though there are several points coincided. For convective rain, there are more points in the "Continental cluster" than the "Maritime cluster", but most points are neither in the "Continental cluster" nor in the "Maritime cluster" and have a tendency to approach the stratiform rain. This indicates that the wet season convective rain in Beijing is neither maritime or continental as described by Bringi et al. (2003), which is likely due to the certain distance between Beijing and the nearest ocean (about

160 km). For stratiform rainfall, the points are more concentrated, even with a wide range of $\log_{10} N_w$ versus $D_m$. More than 85% of the stratiform points appear on the left side of the "stratiform line". The average point of $\log_{10} N_w - D_m$ for all the rainfall events combined (magenta hollow star) also appears on the left side of the "stratiform line" due to the highest population of stratiform in the summer monsoon season (as shown in Table 2). These indicate the lower diameter and higher concentration characteristics of rainfall in Beijing area. The relationship of $\log_{10} N_w - D_0$ (See Fig. S1) shows the line to

classify rain types based on $\log_{10} N_w - D_0$ proposed by Thurai et al. (2016) would misclassify more convective rain as stratiform rain. This is probably because of the complex terrain of Beijing (as shown in Fig. 1a). The high mountain to the west may have substantial impact on the rain evolved from west mainland.

The comparison of DSDs in different parts of China shows interesting results. Even in the same region, the DSDs measured by different instruments have notable differences, such as the differences in Beijing between results from Wen et al. (2017a)

(2DVD, circle) and Tang et al. (2014) (Parsivel, square). In order to reduce the errors caused by different measurement instruments, only DSDs measured by Parsivel disdrometers are analyzed. It is concluded that the east part of China has the lowest mean value of $\log_{10} N_w$ (3.42) with highest mean value of $D_m$ (1.66), while southern China has the highest mean value of $\log_{10} N_w$ (3.86) with middle value of $D_m$ (1.46), and the north part of China has the middle value of $\log_{10} N_w$ (3.60) with


lowest value of $D_m$ (1.15), which indicates that the DSD characteristics are highly correlated to the specific geographical locations and associated climate regimes. The results of Beijing from this study and Chen et al. (2013) show great differences in convective rain and less differences in stratiform rain, which is attributed to different convective systems during different years.

The DSD spectra and $R(D)$ distributions of two rain types are shown in Fig. 6. Substantial differences are observed between these two rainfall types in both DSD spectra and $R(D)$ distributions. The peaks of DSD spectra for both rainfall types are at the same diameter bin around $D \sim 0.5$ mm, while the spectrum for convective is higher than that of stratiform. The peak of $R(D)$ distribution for stratiform rain is at the diameter around 0.9 mm while 1.9 mm for convective rain, which is much larger than where the DSD spectra peaks occur due to the $D^3$ dependency of $R(D)$. In addition, the distribution of $R(D)$ for

convective rain is much lower and broader. The differences in DSD spectra and $R(D)$ distributions indicate that the convective rainfall has a higher concentration of moderate to large size drops, and the large drops contribute more to convective rainfall compared to stratiform rainfall.

### 3.3 DSD characteristics in different rain rate classes

To further understand the characteristics of DSD at different rainfall intensities in Beijing area, the DSD measurements are

divided into 8 classes according to the associated rain rate ($R$): C1, $0.1 \leq R < 0.5$; C2, $0.5 \leq R < 1$; C3, $1 \leq R < 2$; C4, $2 \leq R < 5$; C5, $5 \leq R < 10$; C6, $10 \leq R < 25$; C7, $25 \leq R < 50$; C8, $R \geq 50$ mm h$^{-1}$. Such classification is based on the fact of high frequency of low rain rates in Beijing area, as well as several previous studies including Das and Maitra (2016), Harikumar et al. (2010), Krishna et al. (2016), Sarkar et al. (2015), and Tokay and Short (1996). The DSD sample numbers and rain rate statistics for each category are summarized in Table 3. For each rain rate class, the composite DSD spectrum is

shown in Fig. 7a. Note that almost for all raindrop bins, the concentration of higher rain rate class is higher than that of a lower rain rate class. Furthermore, the breadth of DSD shape increases and the tail of DSD shifts gradually to the larger diameter with the increase of rainfall intensity, which is similar to the previous findings in Taiwan (Seela et al., 2017), south Indian (Jash et al., 2019), Palau (Krishna et al., 2016), and United Kingdom (Islam et al., 2012). All the DSD spectra only have one peak which differs from Krishna et al. (2016) where the spectrum becomes bimodal when rain rate $R > 8$ mm h$^{-1}$. In addition,

the peaks of all DSD spectra are all at a diameter around $D \sim 0.5$ mm, which is different from Jash et al. (2019) for India where the peak position shifts towards larger diameters as rain rate increases.

The mean normalized $R(D)$ of each rain rate class is shown in Fig. 7b, illustrating the contribution of each diameter bin to the total rainwater. The normalized rain rate distributions are unimodal, and the peaks are around $D \sim 0.9$–2.5 mm. The peak position shifts to a larger diameter and the distribution becomes lower and broader as rain rate increases. These results are

similar to those in Jash et al. (2019) for India, but different from those in Peters et al. (2002) for German where the $R(D)$ distribution has a secondary peak at lower rain rate intensity ($R < 1$ mm h$^{-1}$). This analysis implies that raindrops of diameter





0.9–2.5 mm (i.e., moderate size) contribute most towards accumulated rainwater during the rainy season in Beijing area, and the size of drops contributing the most rainfall increases as the rainfall intensity increases.

Variations of normalized intercept parameter ($\log_{10}N_w$) and mass-weighted mean diameter ($D_m$) in each rain rate class are provided in Fig. 8 with box-whisker plot. The central white line and black line in the box represent the median and mean values and the top and bottom lines of the box represent the 75th and 25th percentiles. The top and bottom lines out of the box respectively stand for the 95th and 5th percentiles. It can be seen that $D_m$ values are increasing with the increase of rainfall intensity, while the increasing trend of $\log_{10}N_w$ is not as clear. This could be due to the imbalance between the decrease in small drop concentration and the increase in midsize and large drop concentration at higher rain rate ($R > 10$ mm h$^{-1}$, from C6 to C8). The means and standard deviations of $D_m$, $\log_{10}N_w$, $N_t$, $W$, $\mu$, and $\Lambda$ for each rain rate class are provided in Table 4. Table 4 clearly shows that with the increase of rainfall intensity, the mean values of total number concentration ($N_t$) and liquid water content ($W$) increase, while the mean values of shape parameter ($\mu$) and slope parameter ($\Lambda$) show a decreasing trend, which ensures a wider breadth and lower peak of DSD at high rain rates.

## 3.4 Diurnal variations of DSD characteristics

Since the 1980s, Beijing has been experiencing rapid urbanization, causing a lot of problems among which urban heat island is one of the most well-known phenomena (Yang et al., 2013b). Some studies showed that extreme precipitation events are more likely to occur during the period when the urban heat island (UHI) intensity is high, usually from late afternoon to early morning in Beijing local time (LST) (Li et al., 2008;Song et al., 2014;Yang et al., 2013a;Yang et al., 2017;Zhang et al., 2019b). In order to explore the DSD variations during the day, the diurnal periods are divided into four parts based on the UHI variation described in Yang et al. (2013b): strong UHI stage (S UHI, 2100–0600 LST), weak UHI stage (W UHI, 1100–1600LST), UHI down stage (UHI D, 0600–1100 LST) and UHI up stage (UHI U, 1600–2100 LST). The rain rate and DSD characteristics corresponding to these four stages are shown in Table 5. The DSD spectra and $R(D)$ distributions are shown in Fig 9.

The DSD spectra of different diurnal periods are quite similar to those of different rain rate classes, showing a unimodal shape and peak position at the diameter $D \sim 0.5$ mm. It is notable that the DSD spectra are almost the same at small drop size bins ($D < 1$ mm) and have the same width. As the diameter becomes larger, variations in the DSD spectra start showing up. The DSD spectra of S UHI stage and UHI U stage show similar and higher concentration and the DSD spectra of W UHI stage and UHI D stage have similar but lower concentration, indicating that during the UHI U stage and S UHI stage, high-intensity rainfall is more likely to occur. This is in line with the study in Yang et al. (2017), which showed that the short term high-intensity rainfall was more likely to happen at the UHI U stage and end at the late S UHI stage.

The $R(D)$ distributions for different diurnal periods in Fig. 9b show little difference between UHI U stage and S UHI stage, and distributions at these two stages are lower and broader than the other two stages. For the W UHI stage, the $R(D)$ distribution is the highest and the peak is at diameter around $D \sim 0.9$ mm, and the UHI D stage almost has the same peak around $D \sim 0.9$–1 mm, while the peaks of other two stages are at the diameter around $D \sim 1$ mm. That is, the drop size at the





W UHI stage which contributes most to the accumulated rainwater is smaller than those at UHI U stage or S UHI stage. The box-whisker plots of variation of mass-weighted mean diameter ($D_m$) and normalized intercept parameter ($\log_{10} N_w$) for each diurnal periods show the same results (see Fig. 10). The W UHI stage has the highest mean concentration and the lowest mean $D_m$ value, while the UHI U stage has the largest mean $D_m$ value and the S UHI stage has the lowest mean concentration.

**3.5 DSD characteristics in different months**

To have a better understanding of the seasonal variations of DSD characteristics in Beijing urban area, rain data collected in different months are analyzed. The rain rate and DSD characteristics for different months are shown in Table 6. Figure 11 illustrates the corresponding DSD spectra and $R(D)$ distributions.

As shown in Fig. 11, all the DSD spectra have a peak at diameter $D \sim 0.5$ mm, which are consistent with other classifications
in this study. The DSD in May has a relatively higher concentration while that of July has a relatively lower concentration. At small drop size bins ($D < 1$ mm), the spectra for May and September are similar, while the spectra for other four months are similar. As the diameter increases, the differences between these spectra become larger, and the DSD spectra for July has the highest concentration and October has the lowest concentration. The rainfall with higher concentration and large drops is more likely to happen in July, leading to a high rain rate intensity.

It is also noted that the $R(D)$ distribution for each month is different from each other. The distributions of May, October, and September have a peak at diameter around $D \sim 0.9$ mm, while distributions of June and August have a peak at diameter around $D \sim 1$ mm. The $R(D)$ distribution of July has two peaks at diameter around $D \sim 1$ mm and $D \sim 1.5$ mm. In addition, the $R(D)$ distribution of July is the widest and lowest, suggesting that a wide range of moderate drops contribute mostly to the rain in July. The $D_m$ and $\log_{10} N_w$ in Fig. 12 show an interesting circle of a year: the $D_m$ first goes up and then goes down, while the
$\log_{10} N_w$ goes oppositely, and in July $D_m$ ($\log_{10} N_w$) reaches the highest (lowest) value.

**4 Implications for Radar Rainfall Estimation**

**4.1 Single polarized radar applications**

The power-law relationship between radar reflectivity and rain rate ($Z = aR^b$) is the most widely used for single polarized radar QPE (including the current operational radars in Beijing). However, the coefficient $a$ and exponent $b$ greatly rely on the
DSD variability which may vary in different climate regimes, geographical locations, and rain types (Bringi et al., 2003; Rosenfeld and Ulbrich, 2003; Uijlenhoet, 2001). The default $Z - R$ relationship applied for the operational Weather Surveillance Radar — 1988 Doppler (WSR-88D) systems in the United States is $Z = 300R^{1.4}$ (Fulton et al., 1998), whereas $Z = 200R^{1.6}$ is commonly used in the continental area for stratiform rain (Marshall and Palmer (1948), hereafter referred to as MP-Stratiform relationship). The more appropriate and localized $a$ and $b$ are expected to improve regional radar rainfall





estimation. In the following, the localized $Z - R$ relationships for different rain types are derived, aiming to provide references for operational S-band radar rainfall applications in Beijing.

Figure 13 shows the scatter density plot of rain rate versus horizontal reflectivity, as well as the fitted power-law relations for different rain types. The default NEXRAD algorithm and MP-Stratiform relationship for continental stratiform rain are also

indicated in Fig. 13 for comparison. Figure 13 shows that most of the samples are at low values where both $Z_H$ and $R$ are small, which suggests that the DSD may be under size-controlled conditions (Steiner et al., 2004). The convective relationship $Z = 158R^{1.68}$ has a higher coefficient but lower exponent than the stratiform relation (i.e., $Z = 121R^{2.47}$), which is similar to previous studies. At low reflectivity values ($Z_H < 25$ dBZ), the curve of MP-Stratiform relationship is below that of the local stratiform relation, but at higher values, it reverses. As the mean reflectivity of stratiform rain (21 dBZ) is less than 25 dBZ

(See Table 2), the MP-Stratiform relationship may cause underestimation of rainfall. The default NEXRAD relationship behaves similarly: underestimation at lower reflectivity values and overestimation at higher reflectivity values. Considering the mean reflectivity value of convective rain, the default NEXRAD relationship may cause overestimation of rainfall. In other words, the default relationship $Z = 300R^{1.4}$ should be used with caution for local applications in Beijing.

## 4.2 High frequency (X-band) polarimetric radar applications

A high-resolution dual-polarization X-band radar network is being deployed for urban hydrometeorological applications in Beijing area. To support the radar deployment and facilitate the rainfall applications, the polarimetric parameters, including differential reflectivity $Z_{dr}$ (dB) and specific differential propagation phase shift $K_{dp}$ (°km$^{-1}$) are computed from the DSD measurements. Therein, the T-matrix method (Waterman, 1965) is adopted and the computations are made for X-band radar wavelength. In addition, the polarimetric rainfall relations are derived based on the least-squares method, including $R(K_{dp})$,

$R(K_{dp}, Z_{DR})$, and $R(Z_H, Z_{DR})$. Here $Z_{DR} = 10^{Z_{dr}/10}$ is the differential reflectivity in linear scale. Equations (14)–(17) show the derived X-band radar rainfall relations:

$$R(Z_H) = 0.0304 Z_H^{0.638}, \tag{14}$$

$$R(K_{dp}) = 15.421 K_{dp}^{0.817}, \tag{15}$$

$$R(K_{dp}, Z_{DR}) = 26.778 K_{dp}^{0.946} Z_{DR}^{-1.249}, \tag{16}$$

$$R(Z_H, Z_{DR}) = 4.785 \times 10^{-3} Z_H^{0.978} Z_{DR}^{-3.226}, \tag{17}$$

Previous studies showed that the parameterization errors associated with various radar rainfall relations are among the key factors affecting the derived rainfall performance (Bringi and Chandrasekar, 2001). Here, the parameterization errors in the X-band radar rainfall algorithms are investigated. Figure 14 illustrates the scatter density plots of rain rates derived from $R(Z_H)$, $R(K_{dp})$, $R(K_{dp}, Z_{DR})$, and $R(Z_H, Z_{DR})$ versus the rain rates directly computed from DSD. To quantify the parameterization

errors, the normalized mean absolute error (NMAE) of estimated rainfall rate is calculated, which is defined as:

$$\text{NMAE} = \frac{\langle |R_{EP} - R_D| \rangle}{\langle R_D \rangle}, \tag{18}$$



where the angle brackets stand for sample average; $R_{EP}$ and $R_D$ denote the estimated rain rates derived from parameterized radar rainfall algorithms and DSD information, respectively. The $NMAE_{RR}$ is calculated for different rainfall rate intervals from 0 to 100 mm h$^{-1}$. Figure 15 shows the parameterization error structure of $R(Z_H)$, $R(K_{dp})$, $R(K_{dp}, Z_{DR})$, and $R(Z_H, Z_{DR})$ as a function of rainfall rate.

It can be seen from Fig. 15 that the algorithms based on dual polarization radar parameters can provide better performances than $Z - R$ relationship. In addition, the dual parameter algorithms, namely $R(K_{dp}, Z_{DR})$ and $R(Z_H, Z_{DR})$, have even better performance than the single parameter based algorithm including $R(K_{dp})$. The NMAE has a decreasing trend as the rain rate increase from 1 mm h$^{-1}$ to 60 mm h$^{-1}$. The opposite trend when rain rate is greater than 60 mm h$^{-1}$ may be due to the random errors caused by a few samples of large values. The parameterization errors of $R(K_{dp})$, $R(K_{dp}, Z_{DR})$, and $R(Z_H, Z_{DR})$ become

stable when rain rate is getting higher than 10 mm h$^{-1}$. It is also noted that at low rain rate (less than 1 mm h$^{-1}$), the NMAE of $R(K_{dp})$ is the smallest, while at the rain rate of 1–10 mm h$^{-1}$ , the NMAE of $R(Z_H, Z_{DR})$ is the smallest, and when rain rate is getting higher than 10 mm h$^{-1}$, the NMAE of $R(K_{dp}, Z_{DR})$ becomes the smallest. This again highlights the importance of selecting proper rain rate relations for specific applications.

## 5 Summary and Conclusion

In this paper, 5-year (2014–2018) observations of DSD from a disdrometer deployed at Tsinghua University are analyzed to explore the microphysical characteristics of precipitation during rainy seasons (May–October) in Beijing urban area. The main conclusions are as follows:

1. For all rain events, all the parameters ($D_m$, $D_0$, $D_{max}$, $\log_{10}N_w$, $\log_{10}N_t$, $\log_{10}R$, $\log_{10}\sigma_m$, $\log_{10}T_d$ and $\log_{10}W$) derived from DSD (except $\sigma_m$) have a positive skewness, indicating a high frequency of low values and low frequency of high

values in Beijing urban area. And more than half of the DSD measurements are characterized by rainfall rate less than 1 mm h$^{-1}$.

2. The mean values of $\log_{10}N_w$ and $D_m$ of convective rain are higher than that of stratiform rain, indicating a higher raindrop concentration and larger drop size during convective events. This is also in line with the raindrop spectra and normalized $R(D)$ distribution. In addition, $\log_{10}N_w$ of convective rain is negatively skewed, which is opposite to that of

stratiform rain. For both types of rain, the $D_m$ values are higher but the distributions are narrower at higher rainfall intensities.

3. There is a clear boundary to distinguish between convective and stratiform rain from the scatterplot of $\log_{10}N_w$ versus $D_m$. However, the convective rain in Beijing area is neither continental or maritime as described by Bringi et al. (2003), due to the particular location and complex topography. Moreover, the comparison with results in different parts of China

shows that the DSD variability is closely related to geographic location, climate regimes and study periods.



4.  Stratified by rain rate, the DSD spectra become higher and wider as the rain rate increases, but all have peaks at the same diameter D ~ 0.5 mm. The peaks of the normalized $R(D)$ distribution move to larger diameter size (still within the midsize range) and the distribution becomes lower and wider as the rain rate increases. Meanwhile, the $D_m$ and $\log_{10}N_w$ show an increasing trend while the slope parameter ($\mu$) shows a decreasing trend as the rain rate increases.

5.  During the periods of strong UHI and UHI up stages, the DSD spectra trend to have a higher concentration at large size drops, and larger $D_m$ values than other periods, indicating intense rainfall at these periods. The DSD has similar characteristics in July and August. In addition, the $D_m$ and $\log_{10}N_w$ values show a diurnal cycle and an annual cycle. All these findings indicate great temporal variabilities of DSD in Beijing.

6.  The localized $Z - R$ relationship derived from DSD is quite different from the operational NEXRAD algorithm that may underestimate (overestimate) rainfall at low (high) rain intensity. The error structures of different algorithms show that the polarimetric radar rainfall relations $R(K_{dp})$, $R(K_{dp}, Z_{DR})$, and $R(Z_H, Z_{DR})$ have greater potential than $Z - R$ methods for urban QPE.

The statistical analysis of DSD characteristics presented in this study not only provides a further understanding of precipitation microphysical variabilities in Beijing but also provides indications for future model development to improve local precipitation forecast. In addition, a high-resolution X-band dual polarization radar network is being deployed in Beijing. This study is expected to provide references for future development of localized radar rainfall algorithms. Nevertheless, the DSD spectra also show the limitations of Parisvel[2] disdrometer in measuring small raindrops. Future study should be carried out with multiple instruments including a two-dimensional video disdrometer just deployed in this area. In addition, further investigation on the spatial variability of DSD induced by the complex micro-topography in urban area should be conducted in a future study.

**Acknowledgments**

This research was supported by the Ministry of Science and Technology of the People's Republic of China under grant 2013DFG72270 and National Key Research and Development Program of China under grant 2018YFA0606002. Yu Ma was also supported by the China Scholarship Council and Tsinghua University Tutor Research Fund. Participation of V. Chandrasekar and Haonan Chen were also supported by the U.S. National Science Foundation Hazards SEES program and California Department of Water Resources, respectively.

**Supplementary Materials**

Figure S1: Scatter density plot of $log_{10}N_w$ versus $D_0$: (a) the total rainfall events; (c) stratiform events; (d) convective events. (b) is the scatterplot of $log_{10}N_w$ versus $D_0$ for convective (red circle dots) and stratiform (blue square dots) cases. The black dashed line is the $\log_{10}N_w - D_0$ relationship for stratiform rain reported by Thurai et al. (2016).





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





**Tables:**

**Table 1: Statistics of DSD parameters for all observations: $D_m$, $D_0$, $D_{max}$, $\log_{10}N_w$, $\log_{10}N_t$, $\log_{10}R$, $\log_{10}\sigma_m$, $\log_{10}T_d$ and $\log_{10}W$**

| Parameters | $D_m$ (mm) | $D_0$ (mm) | $D_{max}$ (mm) | $\log_{10}N_w$ ($N_w$ in m$^{-3}$ mm$^{-1}$) | $\log_{10}N_t$ ($N_t$ in m$^{-3}$) | $\log_{10}R$ ($R$ in mm h$^{-1}$) | $\log_{10}\sigma_m$ ($\sigma_m$ in mm) | $\log_{10}T_d$ - | $\log_{10}W$ ($W$ in g m$^{-3}$) |
|---|---|---|---|---|---|---|---|---|---|
| Min | 0.376 | 0.304 | 0.687 | 0.435 | 0.747 | -1.000 | -1.071 | 1.041 | -2.244 |
| Media | 1.054 | 0.949 | 1.875 | 3.596 | 2.301 | -0.134 | -0.517 | 2.253 | -1.277 |
| Mean | 1.148 | 1.037 | 1.987 | 3.595 | 2.311 | -0.070 | -0.521 | 2.264 | -1.229 |
| Max | 5.546 | 6.777 | 7.500 | 5.669 | 3.798 | 2.037 | 0.064 | 3.739 | 0.678 |
| SD | 0.456 | 0.431 | 0.913 | 0.621 | 0.476 | 0.558 | 0.126 | 0.450 | 0.495 |
| Skewness | 1.780 | 2.115 | 1.550 | 0.040 | 0.058 | 0.648 | -0.140 | 0.107 | 0.571 |
| Kurtosis | 9.010 | 12.252 | 6.535 | 4.070 | 2.859 | 3.150 | 3.121 | 2.711 | 3.074 |

**Table 2: Statistical properties of DSD parameters for convective and stratiform rain**

| Parameters | Number - | $D_m$ (mm) | $N_w$ (m$^{-3}$ mm$^{-1}$) | $N_t$ (m$^{-3}$) | $R$ (mm h$^{-1}$) | $\sigma_m$ (mm) | Td - | $W$ (g m$^{-3}$) | $R_H$ (dBZ) | $Z_{dr}$ (dB) | $K_{dp}$ (° km$^{-1}$) |
|---|---|---|---|---|---|---|---|---|---|---|---|
| Convective | 3650 | 1.909 | 4570 | 1042 | 16.2 | 0.385 | 1024 | 0.745 | 40.227 | 1.579 | 1.113 |
| Stratiform | 39968 | 1.078 | 3881 | 312 | 1.1 | 0.308 | 250 | 0.072 | 21.052 | 0.421 | 0.037 |
| Total | 43618 | 1.148 | 3938 | 373 | 2.4 | 0.314 | 315 | 0.128 | 22.656 | 0.518 | 0.128 |

**Table 3: Number and DSD retrieved rain rate statistics of each rain rate class**

| | Rain Rate Threshold | No. of Samples | Mean mm h$^{-1}$ | SD mm h$^{-1}$ | Skewness | Kurtosis |
|---|---|---|---|---|---|---|
| C1 | $0.1 \leq R < 0.5$ | 16464 | 0.27 | 0.11 | 0.36 | 1.96 |
| C2 | $0.5 \leq R < 1$ | 9340 | 0.72 | 0.14 | 0.29 | 1.92 |
| C3 | $1 \leq R < 2$ | 7466 | 1.43 | 0.29 | 0.29 | 1.90 |
| C4 | $2 \leq R < 5$ | 6145 | 3.08 | 0.82 | 0.62 | 2.26 |
| C5 | $5 \leq R < 10$ | 2141 | 6.93 | 1.41 | 0.47 | 2.06 |
| C6 | $10 \leq R < 25$ | 1463 | 15.47 | 4.11 | 0.58 | 2.25 |
| C7 | $25 \leq R < 50$ | 446 | 34.85 | 6.91 | 0.42 | 1.96 |
| C8 | $R \geq 50$ | 153 | 62.98 | 10.95 | 1.39 | 5.44 |



**Table 4: Number and DSD retrieved rain rate statistics of each rain rate class**

|  | $D_m$ (mm) | | $\log_{10}N_w$ (m$^{-3}$ mm$^{-1}$) | | $N_t$ (m$^{-3}$) | | $W$ (g m$^{-3}$) | | $\mu$ | | $\Lambda$ | |
|---|---|---|---|---|---|---|---|---|---|---|---|---|
|  | Mean | SD | Mean | SD | Mean | SD | Mean | SD | Mean | SD | Mean | SD |
| C1 | 0.91 | 0.27 | 3.47 | 0.64 | 177.06 | 261.50 | 0.02 | 0.01 | 12.40 | 10.09 | 20.90 | 16.54 |
| C2 | 1.06 | 0.32 | 3.62 | 0.63 | 304.90 | 429.05 | 0.05 | 0.02 | 9.05 | 7.90 | 14.63 | 12.87 |
| C3 | 1.20 | 0.37 | 3.68 | 0.60 | 392.99 | 474.86 | 0.09 | 0.03 | 7.00 | 6.23 | 10.86 | 9.23 |
| C4 | 1.37 | 0.43 | 3.73 | 0.59 | 547.35 | 514.13 | 0.18 | 0.06 | 5.55 | 5.16 | 8.19 | 6.52 |
| C5 | 1.64 | 0.51 | 3.70 | 0.55 | 693.45 | 421.56 | 0.36 | 0.08 | 4.65 | 4.34 | 6.05 | 4.26 |
| C6 | 2.01 | 0.56 | 3.62 | 0.50 | 947.33 | 447.56 | 0.71 | 0.19 | 3.06 | 2.59 | 3.93 | 2.18 |
| C7 | 2.25 | 0.36 | 3.72 | 0.32 | 1886.51 | 866.64 | 1.50 | 0.30 | 1.46 | 1.53 | 2.51 | 0.91 |
| C8 | 2.32 | 0.19 | 3.90 | 0.20 | 3240.38 | 1012.48 | 2.68 | 0.48 | 0.62 | 0.79 | 2.01 | 0.44 |

**Table 5: Mean and Standard Deviation (SD) Values of $R$, $D_m$, $\log_{10}N_w$, $N_t$, $W$, $\mu$, and $\Lambda$ for different diurnal periods based on UHI intensity**

|  | $R$(mm h$^{-1}$) | | $D_m$ (mm) | | $\log_{10}N_w$ (m$^{-3}$ mm$^{-1}$) | | $N_t$ (m$^{-3}$) | | $W$ (g m$^{-3}$) | | $\mu$ | | $\Lambda$ | |
|---|---|---|---|---|---|---|---|---|---|---|---|---|---|---|
|  | Mean | SD | Mean | SD | Mean | SD | Mean | SD | Mean | SD | Mean | SD | Mean | SD |
| W UHI | 1.88 | 4.31 | 1.11 | 0.42 | 3.59 | 0.60 | 342.15 | 499.30 | 0.10 | 0.19 | 15.06 | 13.63 | 9.32 | 8.49 |
| UHI U | 2.04 | 4.10 | 1.10 | 0.41 | 3.70 | 0.58 | 378.44 | 398.08 | 0.12 | 0.18 | 15.27 | 14.48 | 9.33 | 8.90 |
| S UHI | 2.82 | 6.94 | 1.18 | 0.51 | 3.57 | 0.65 | 380.88 | 488.27 | 0.15 | 0.30 | 14.09 | 13.45 | 8.78 | 8.45 |
| UHI D | 2.60 | 6.79 | 1.18 | 0.46 | 3.56 | 0.64 | 385.00 | 563.30 | 0.14 | 0.30 | 13.97 | 13.95 | 8.61 | 8.43 |

**Table 6: Mean and Standard Deviation (SD) Values of $R$, $D_m$, $log_{10}N_w$, $N_t$, $W$, $\mu$, and $\Lambda$ for each month**

|  | $R$(mm h$^{-1}$) | | $D_m$ (mm) | | $\log_{10}N_w$ (m$^{-3}$ mm$^{-1}$) | | $N_t$ (m$^{-3}$) | | $W$ (g m$^{-3}$) | | $\mu$ | | $\Lambda$ | |
|---|---|---|---|---|---|---|---|---|---|---|---|---|---|---|
|  | Mean | SD | Mean | SD | Mean | SD | Mean | SD | Mean | SD | Mean | SD | Mean | SD |
| May | 1.34 | 2.09 | 1.04 | 0.39 | 3.74 | 0.68 | 440.30 | 602.46 | 0.08 | 0.10 | 9.20 | 8.05 | 16.44 | 16.19 |
| Jun | 2.10 | 4.61 | 1.16 | 0.47 | 3.55 | 0.66 | 363.01 | 464.13 | 0.12 | 0.21 | 8.61 | 8.09 | 13.83 | 12.76 |
| Jul | 3.61 | 8.20 | 1.28 | 0.50 | 3.49 | 0.58 | 358.84 | 507.50 | 0.18 | 0.36 | 8.34 | 9.34 | 12.53 | 12.58 |
| Aug | 2.80 | 6.74 | 1.16 | 0.45 | 3.57 | 0.62 | 375.65 | 476.69 | 0.15 | 0.29 | 9.70 | 9.60 | 15.03 | 14.80 |
| Sep | 1.63 | 4.10 | 1.04 | 0.42 | 3.70 | 0.64 | 418.63 | 612.39 | 0.10 | 0.18 | 10.29 | 9.35 | 17.19 | 15.26 |
| Oct | 1.07 | 1.37 | 1.03 | 0.34 | 3.68 | 0.55 | 307.38 | 312.11 | 0.07 | 0.07 | 7.82 | 6.86 | 14.14 | 12.38 |



**Figures:**

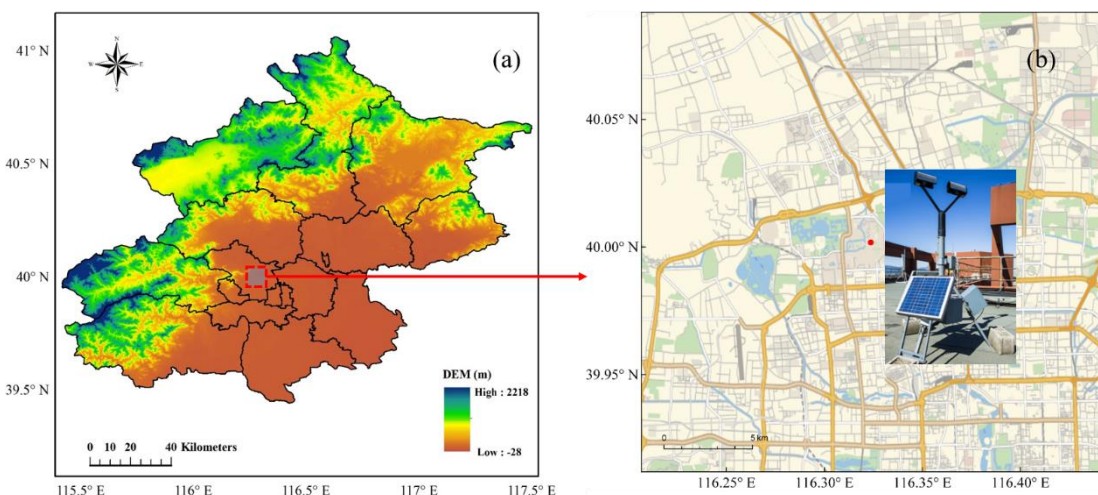

**Figure 1: The topography of Beijing and the location of Parsivel$^2$ disdrometer deployed at Tsinghua University campus.**



**Figure 2: Histograms of different DSD parameters for all selected rainfall: (a) the mass-weighted mean diameter, $D_m$ (mm); (b) median volume diameter $D_0$ (mm); (c) maximum diameter, $D_{max}$ (mm); (d) generalized intercept parameter, $\log_{10} N_w$ ($N_w$ in m⁻³ mm⁻¹); (e) total number concentration, $\log_{10} N_t$ ($N_t$ in m⁻³); (f) rain rate $\log_{10} R$ ($R$ in mm h⁻¹); (g) mass spectrum standard deviation $\log_{10} \sigma_m$ ($\sigma_m$ in mm); (h) total number of rain drops $\log_{10} T_d$, (i) liquid water content $\log_{10} W$ ($W$ in g m⁻³).**

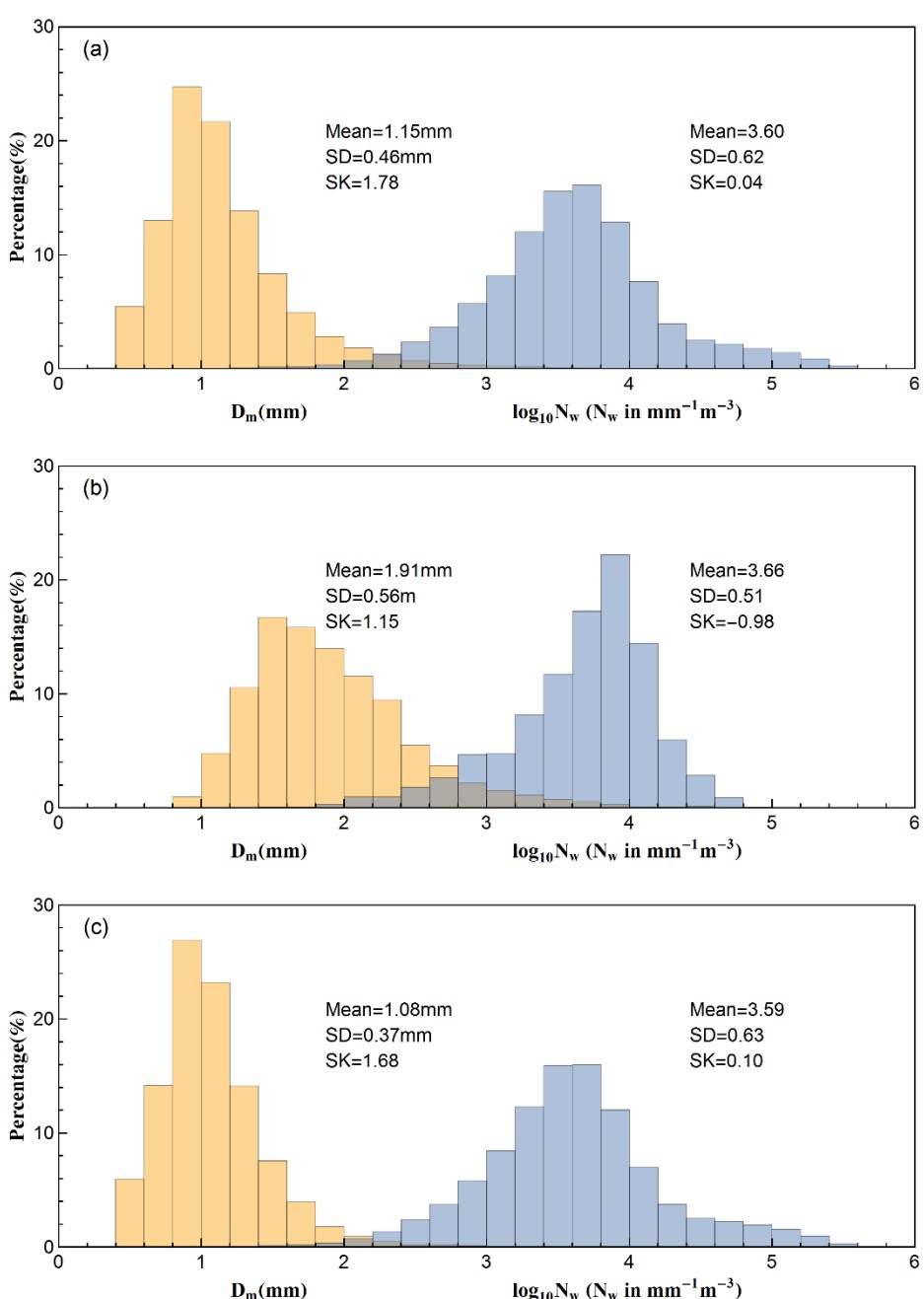

**Figure 3: Histograms of $D_m$ and $\log_{10} N_w$ for (a) all the rainfall events, (b) convective events, and (c) stratiform events. Mean values, standard deviation (SD), and skewness (SK) are also shown in the respective panels.**





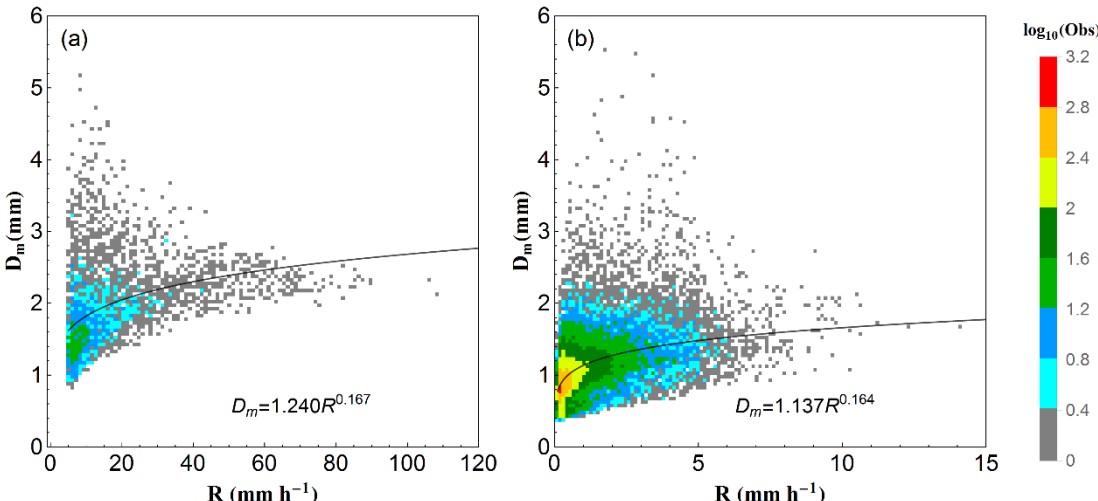

**Figure 4: Scatter density plot for $D_m$ (mm) versus $R$ (mm h$^{-1}$) for (a) convective events, (b) stratiform events. The fitted power-law relationships are also provided in each panel adopting a least-squares method.**

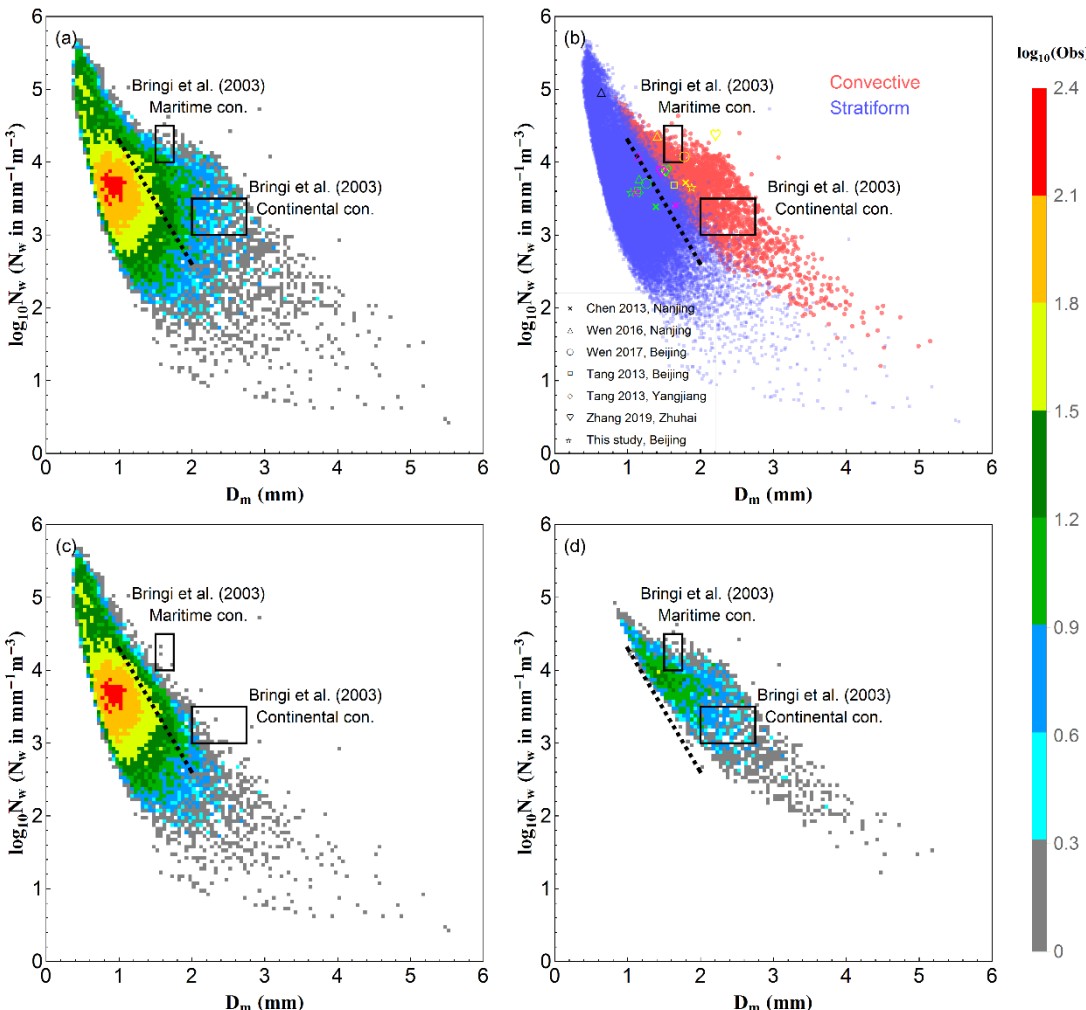

**Figure 5: Scatter density plot of $\log_{10}N_w$ versus $D_m$: (a) the total rainfall events; (c) stratiform events; (d) convective events. (b) is the scatterplot of $\log_{10}N_w$ versus $D_m$ for convective (red circle dots) and stratiform (blue square dots) cases. The two grey rectangles in each subplot correspond to the maritime and continental convective clusters, and the black dashed line is the $\log_{10}N_w - D_m$ relationship for stratiform rain reported by Bringi et al. (2003). The cross, hollow triangles, squares, diamonds, and hearts in (b) represent the averaged values obtained in previous studies for different parts of China The colors of these symbols present different events: magenta for total rainfall events; green for convective events; yellow for stratiform events, and black for the shallow events.**





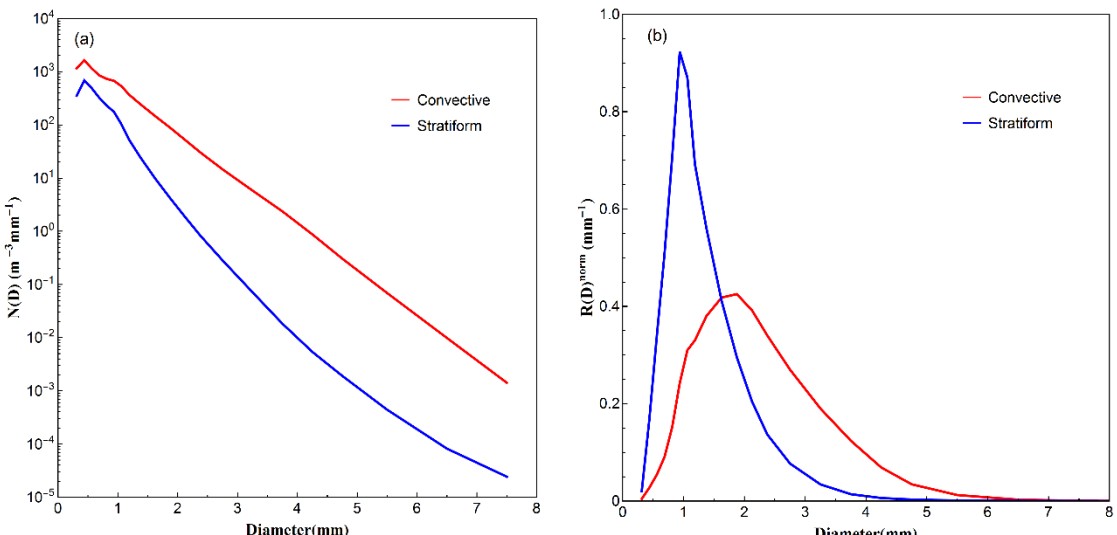

**Figure 6: Composite raindrop spectra (a) and normalized $R(D)$ distributions (b) for different rain types.**

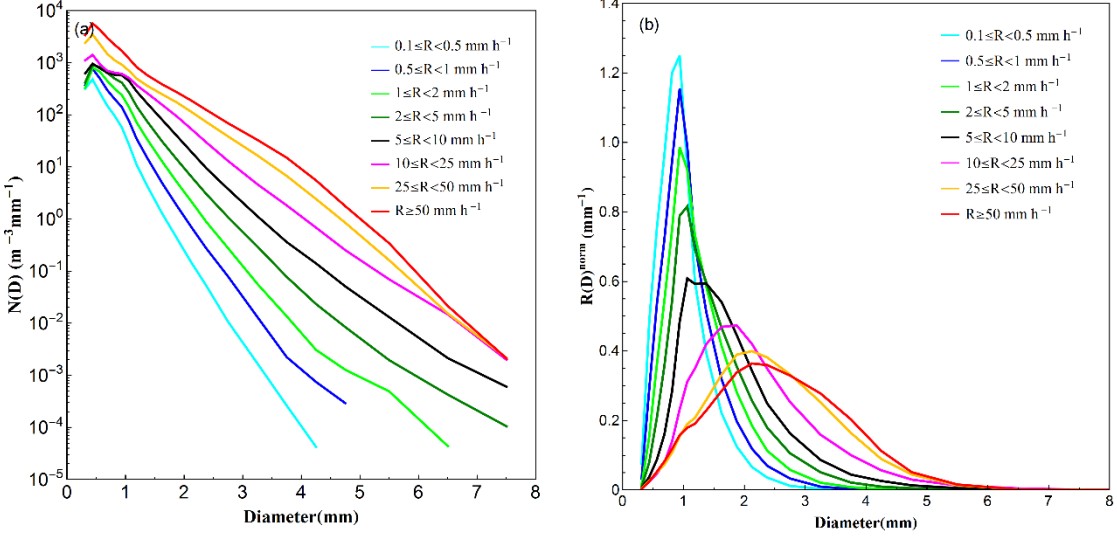

**Figure 7: Same as Figure 6, but for different rain rate classes.**





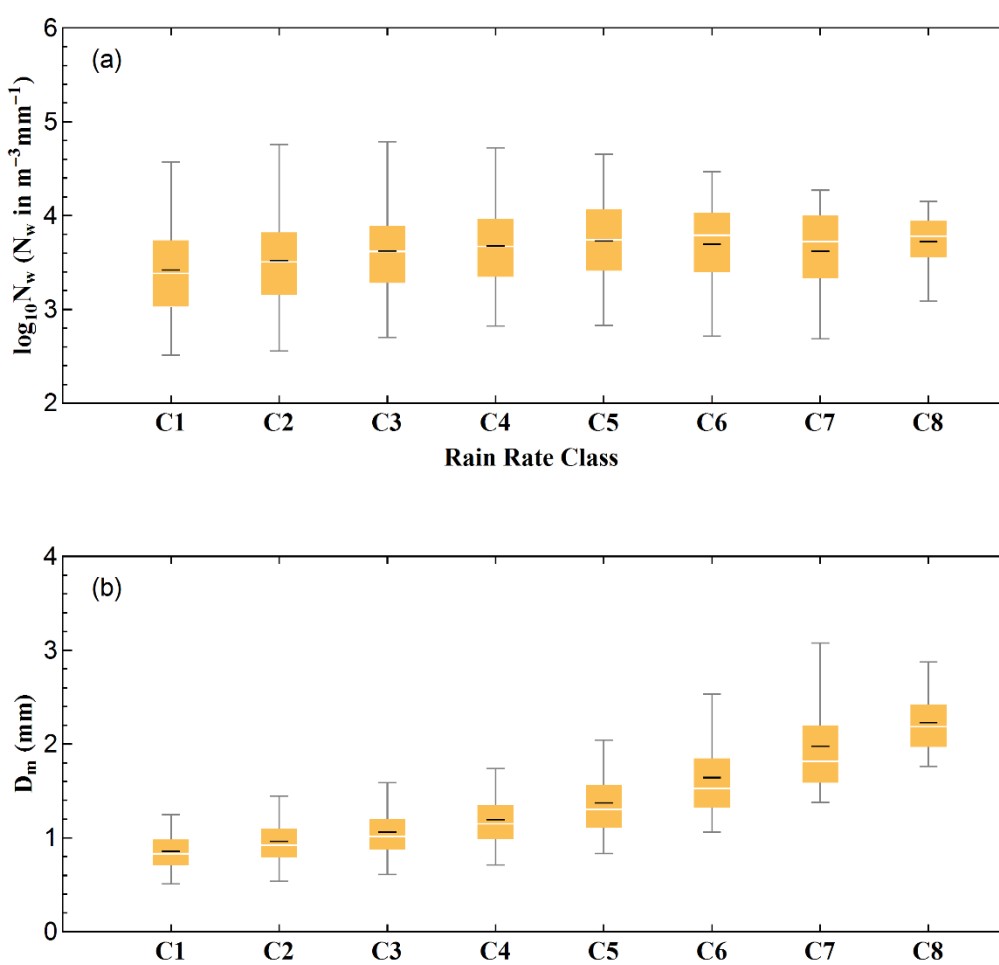

**Figure 8: Variation of normalized intercept parameter, $\log_{10} N_w$ (a) and mass-weighted mean diameter, $D_m$ (b) for different rain rate classes. The white central line of the box indicates the median, the black central line in the box indicates the mean values, and the bottom and top lines of the box indicate the 25th and 75th percentiles, respectively. The bottom and top of the dashed vertical lines indicate the 5th and 95th percentiles, respectively.**





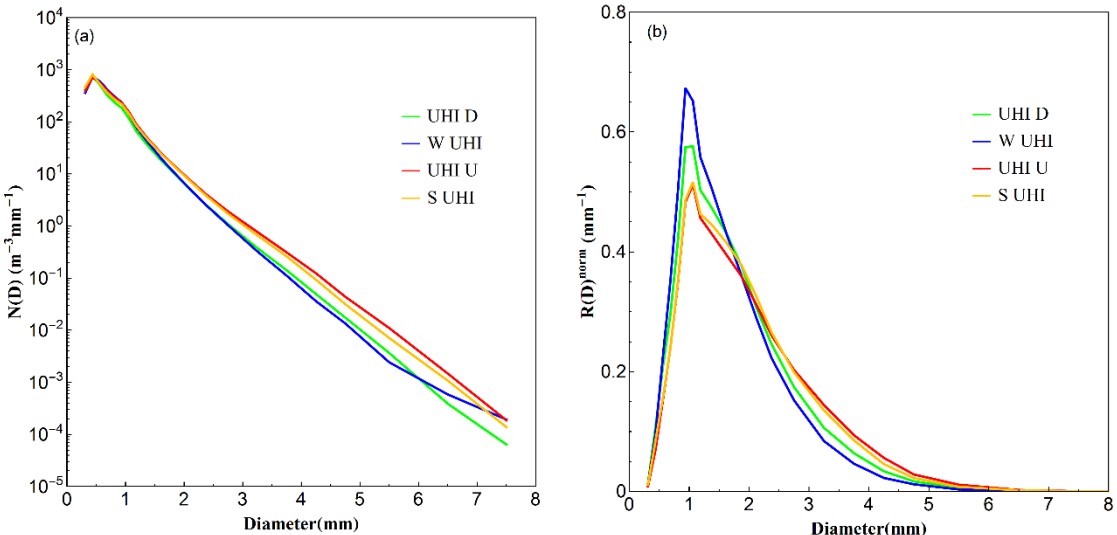

Figure 9: Same as Figure 6, but for different diurnal periods based on UHI intensity.





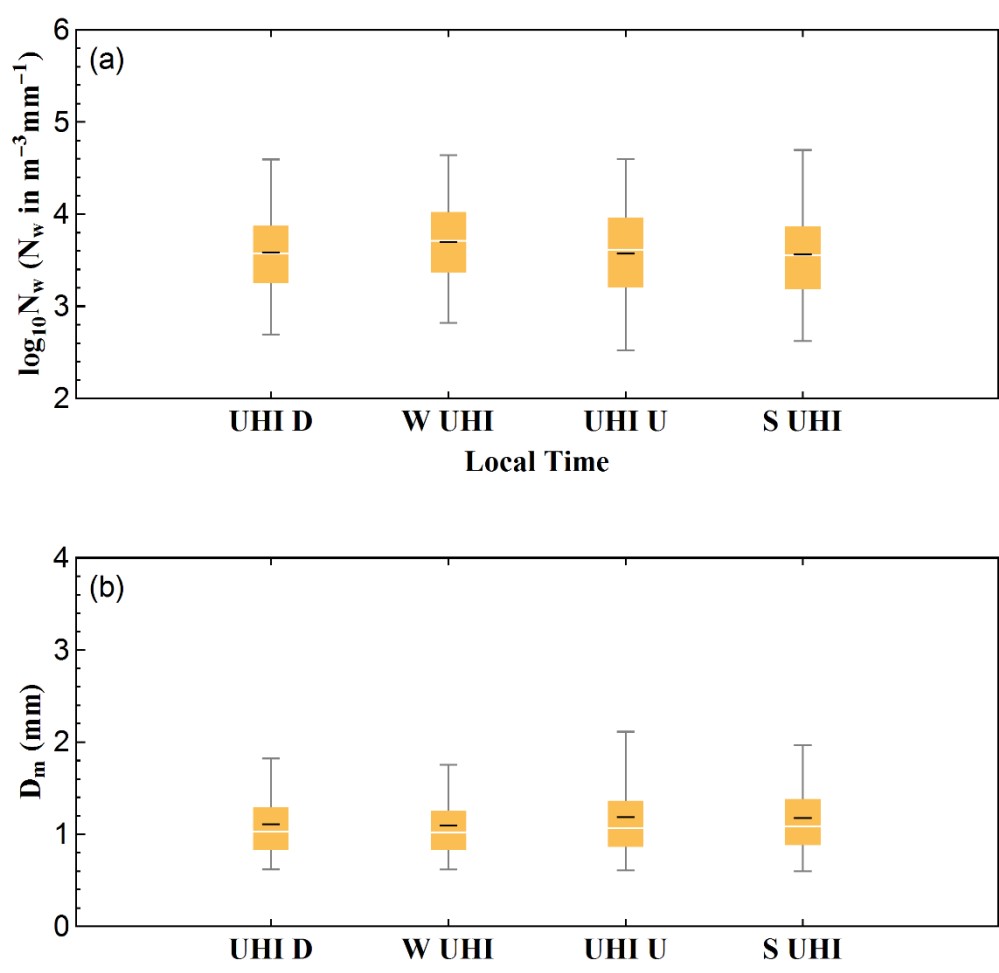

**Figure 10: Same as Figure 8, but for different diurnal periods based on UHI intensity.**





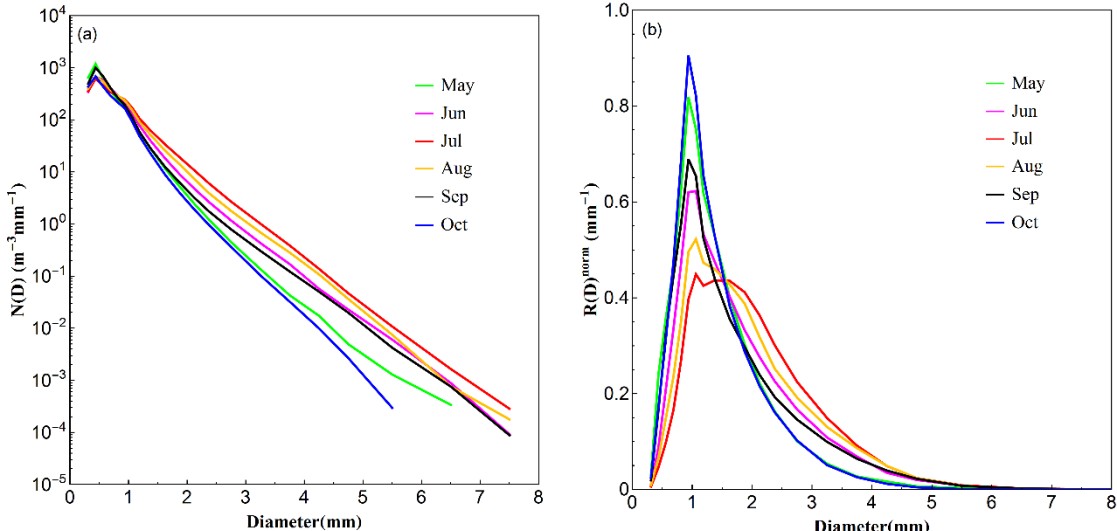

**Figure 11: Same as Figure 6, but for different months.**





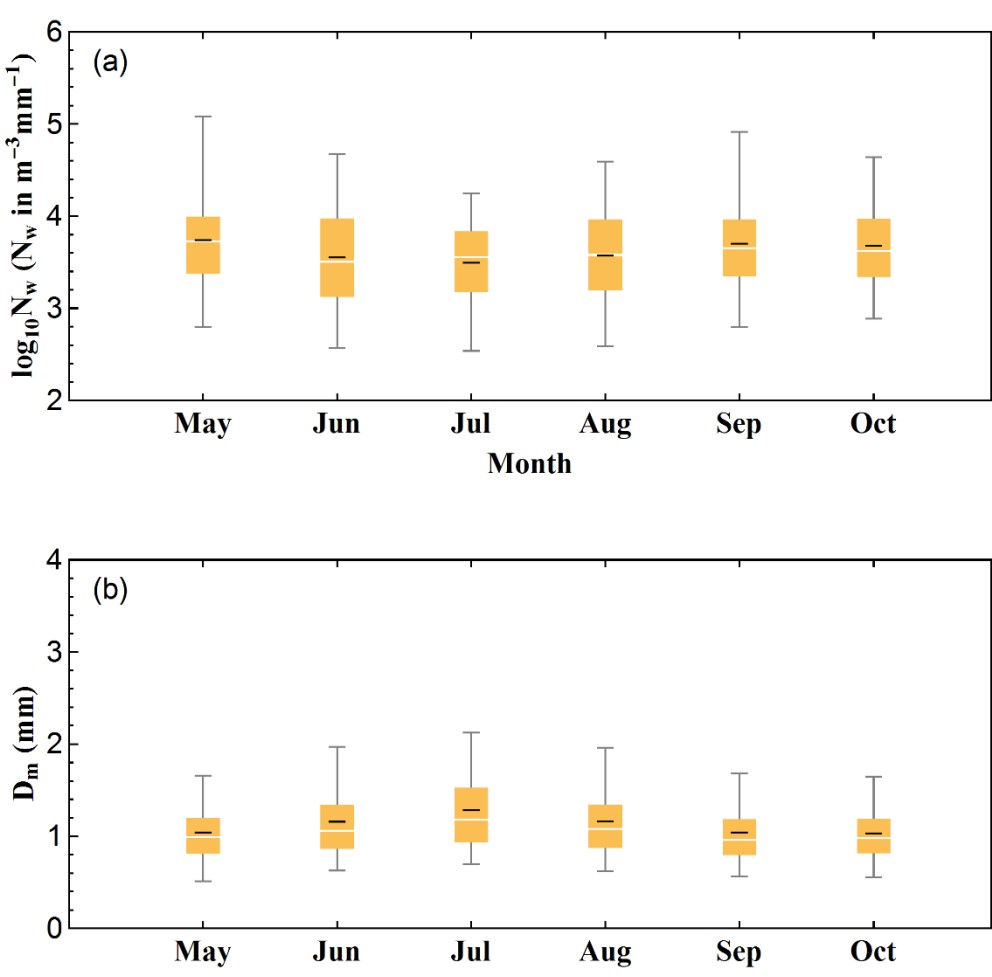

Figure 12: Same as Figure 8, but for different months.





**Figure 13: Scatter density plot of $R$ (mm h$^{-1}$) versus $Z_H$ (dBZ) for all rain events. The black, red, and blue curves respectively stand for the fitted power-law relations for total rain, convective rain, and stratiform rain. The purple and green dashed lines denote the default NEXRAD $Z - R$ relation (Fulton et al., 1998) and a commonly used continental stratiform rain relation (Marshall and Palmer, 1948), respectively.**





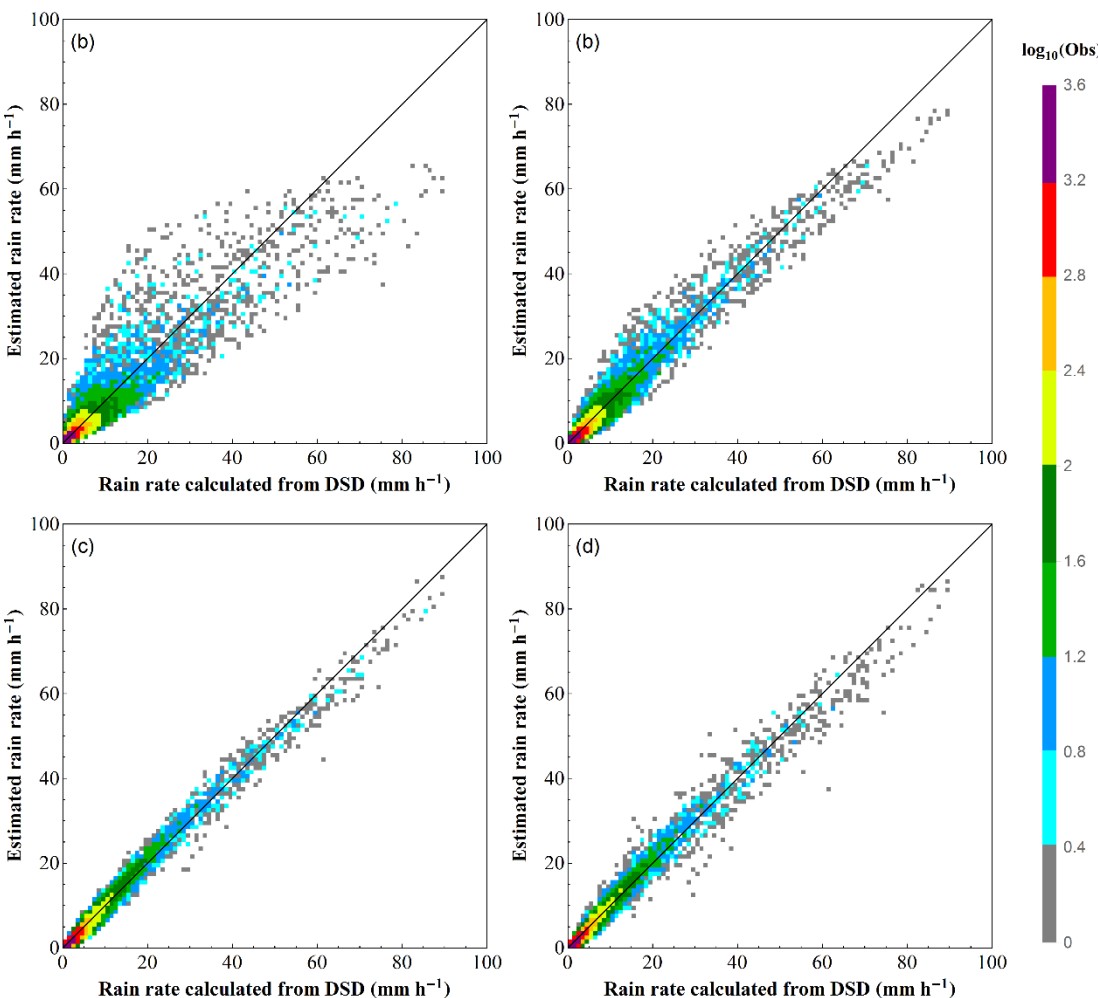

**Figure 14:** Scatter density plots of rainfall rates estimated from radar rainfall relations versus rain rates calculated directly from DSD: (a) $R(Z_H)$, (b) $R(K_{dp})$, (c) $(K_{dp}, Z_{DR})$, and (d) $R(Z_H, Z_{DR})$. The black diagonal line in each panel represents the 1–1 relationship.





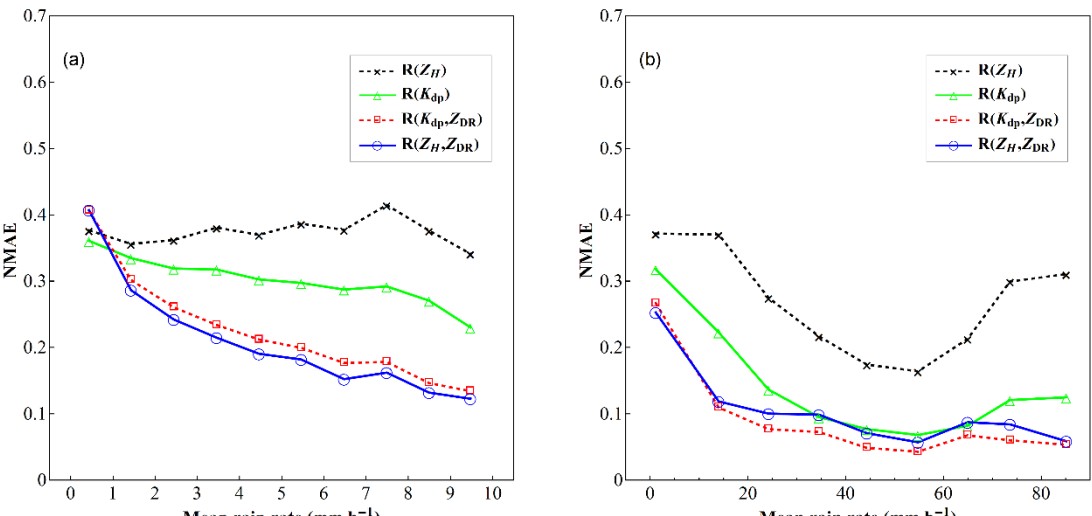

**Figure 15: Parameterization error structure of $R(Z_H)$, $R(K_{dp})$, $R(K_{dp}, Z_{DR})$, and $R(Z_H, Z_{DR})$ as a function of rainfall rate: (a) for mean rain rate less than 10 mm h$^{-1}$; (b) for rain rate of the whole dataset.**