# Peer review of "Figure S1: Scatter density plot of $\log_{10} N_w$ versus $D_0$ : (a) the total rainfall events; (c) stratiform events; (d) convective events. (b) is the scatterplot of $\log_{10} N_w$ versus $D_0$ for convective (red circle dots) and stratiform (blue square dots) cases. The black dashed line"

_Hydrology and Earth System Sciences, 2019_

## Referee Comment (RC1) · Anonymous Referee #1 · 22 Jun 2019

Comments on "Statistical characteristics of raindrop size distribution during rainy seasons in Beijing urban area and implications for radar rainfall estimation", by Ma et al., submitted to Hydrology and Earth System Sciences.

The authors present a well-designed study of DSD over a dense urban area. The results can advance our understanding of rainfall microphysics and improve radar QPE in urban areas. There are some places in the manuscript that need further clarification, but other than that, this is a well-written paper and can be accepted after revision. My specific comments are listed below (not necessarily in order of importance).

Specific comments:
1)  Please explain the meaning of $\log_{10}N_w$ and $D_m$ on their first occurrence.
2)  The Introduction needs to be further strengthened. It seems that this study only differs from previous studies simply through using a long-term dataset, as can be inferred from the current version, which is actually not.
3)  I would suggest not to mention "local microphysics" in P2, Line 4, as apparently this present study does not provide much interpretation of rainfall microphysics. The main objective is for better characterizations of DSD in urban region and potential improvement for radar QPE.
4)  P1, Line 21, what does "UHI up stage of a day" mean? Please clarify.
5)  Since there is a dual-pol radar collocated with the disdrometer, I wonder how the dual-pol radar fields are utilized in this study. The dual-pol fields used in this study are simulated using the T-matrix method. How accurate is the simulation?
6)  Hail contamination remains a challenge for radar QPE. However, this is how dual-pol radar can surpass conventional radar (using the KDP field). It seems strange to me that the authors remove hail from all their records, as this will degrade the significance of their study. Please justify.
7)  The threshold of 5 mm/h for separating convective and stratiform rainfall is small compared to previous studies. Please justify.
8)  Please remove the texts P7, Lines 9-12. They can be moved to the caption of Figure 5. Similarly for P9, Lines 4-6.
9)  Figure 5, caption, what does "shallow events" mean? Please explain.
10) Figure 5 and texts, I'm not sure if it is reasonable to compare this study with previous studies, as clearly this study present climatological features of DSD, while the referenced studies seem to be event-based.
11) I would suggest to present frequency distribution of rain rates among different UHI stages, along with DSD parameters in Figure 9. As the authors explained differences of DSD parameters for different rain rates in previous section, differences of DSD parameters among UHI stages might be simply due to rain rate differences. This suggestion also applies for the analysis of seasonal cycle in section 3.5.
12) Grammar and wording need double check. There are some typos throughout the manuscript, for instance, "P1,Line 34, warn should be warm", etc.

Long Y.

---

## Referee Comment (RC2) · Anonymous Referee #2 · 2 Aug 2019

General Comments

In summary, this study analyzed the statistical characteristics of raindrop size distribution (DSD) during rainy seasons (May-October) in Beijing based on a 5-year observation (2014-2018) from a Parsivel2 disdrometer deployed at Tsinghua University, compared the differences in diameter and concentration between rain types, rainfall intensity, urban heat island (UHI) stages and months, and finally explored its implications for two types of radar rainfall estimations. The manuscript is overall detailed and well written with analysis of DSD parameters and suggestions for precipitation forecast, while it has some minor problems and lacks further explanation of precipitation microphysics. Therefore, I suggest a minor revision and encourage the authors to improve this manuscript. Detailed suggestions are listed below. As I'm not working on this specific researching area, some suggestions may not be suitable for this manuscript, and the authors can decide whether or not to accept them.

Major Comments

(1) I've noticed the authors actually show their results together with discussions in Section 3 and 4, while I personally prefer an independent Discussion Section to clarify the differences and significance of this study compared to others on DSD characteristics in Beijing (and other cities). For example, the authors derived an opposed conclusion referred to Wen and Zhang's work (P7 L10), and it would be better if the authors mark their observation locations in Fig. 1(b), explain the differences in physical mechanism and show detailed possible causes.

(2) Abstract Section. I suggest the authors should first clarify the meanings before using symbols or abbreviations such as Dm and lgNw when showing results in Abstract Section. In addition, although P4 L15 defined Nw as "normalized intercept parameter", I have not found its clear physical meaning which expected to be similar to Nt, the total number concentration.

(3) P6, L15, the authors use specific mean and standard derivation values of rain rate (R) as thresholds to separate convective rain from stratiform rain. However, it seems that R is only related to D spectra considering equation (10) and (3), so in my opinion this classification method is equivalent to solving nonlinear equations and will probably cause the "clear boundary" in DSD characteristics between rain types mentioned in Abstract Section. The authors should pay attention to the classification method chosen in this study, and it would be better if they obtain more information on rain types from other data sources.

(4) There is a mistake in Table 5. The correct UHI stage labels in the table should be UHI D, W UHI, UHI U and S UHI, which is consistent with Figure 9 and 10 indicating

UHI W stage has the largest mean concentration and lowest Dm.

(5) Figure 13. This figure may mislead the readers as the study focused mainly on low rain rate values (less than 25 mm/h). I suggest the authors should plot it on double logarithmic coordinates, which will make it a linear relationship (i.e. convert $Z=238R^{1.57}$ to $lgZ=1.57lgR+lg238$). Besides, the derived line for total rainfall are below both convective and stratiform lines for low rain rate values, and the authors should explain this.

(6) Section 4.1 and 4.2. How did the authors figure out the relationship equations (14)-(17)? In my opinion, it is more likely that the uncertainty in parameter values, other than suitability of algorithms, may be the main sources of normalized absolute error (NMAE).

Minor Comments

(7) P2, L19-26. These sentences are weird to read with duplicate words such as "high spatial and temporal variabilities". I guess the authors here wanted to elaborate the complexity of measuring and modeling precipitation in Beijing due to its high urbanization (i.e. densely populated) and large heterogeneity (i.e. high spatial and temporal variabilities), and show the significance of analyzing DSD characteristics which could help us to understand urban precipitation. I suggest that the authors should rewrite this part to keep it concise and clear.

(8) P2, L21, "... stations network de Vos et al., 2017", add "by" after "network". In addition, I prefer a standard usage of references in the text.

(9) P2, L22, "monitoring networks . . . have been applied", here using "established" may be a better choice.

(10) P2, L34, "warn" -> "warm".

(11) P3, L5, "methodologies" -> "methods".

[Figure]

(12) P3, L7. I suggest the word "Section" should be capitalized.

(13) P3, L15-17 and L25. From the manuscript, I guess these 32 non-uniform bins are set by THUD and fixed for all rainfall events, leaving the maximum observable diameter to be 24.5 mm. However, P5 L20 mentioned that the biggest raindrops ever reported are around 8 mm. The authors should clearly point it out if the latter diameter value can only represent precipitation in Beijing.

(14) P3, L24-25. How to obtain Dj if only the number of raindrops belonging to each bin was recorded? I've noticed that the maximum value of Dmax happened to be 7.5 mm in Table 1, so I guess there should exists a bin ranging from 7 mm to 8 mm, and the authors took its average as corresponding diameter.

(15) P5, L30. How did the authors figure out the relationship between Dm and D0?

(16) P13, L9. I guess the authors missed "(MP-Strariform)" after "NEXRAD".

---

## Author Comment (AC1) · 21 Aug 2019

We thank the reviewer for the kind words. We appreciate all the valuable comments and suggestions provided by the reviewer. We have carefully revised this manuscript based on the reviewer's comments. Detailed responses are attached in a separate document.

Please also note the supplement to this comment: https://www.hydrol-earth-syst-sci-discuss.net/hess-2019-210/hess-2019-210-AC1-supplement.pdf

210, 2019.

**Supplement:**

**Response to Reviewer #1**

*Overall comments:*
*The authors present a well-designed study of DSD over a dense urban area. The results can advance our understanding of rainfall microphysics and improve radar QPE in urban areas. There are some places in the manuscript that need further clarification, but other than that, this is a well-written paper and can be accepted after revision. My specific comments are listed below (not necessarily in order of importance).*
**Response:** We thank the reviewer for the kind words. We appreciate the reviewer's time and effort spent on our manuscript. We have carefully revised this manuscript based on the reviewer's comments. In the text below we quote the reviewers' comments verbatim and we follow them with our detailed responses in red.

*Specific comments:*
*1. Please explain the meaning of $log_{10}N_w$ and $D_m$ on their first occurrence.*
**Response:** We thank the reviewer for this suggestion. $log_{10}N_w$ is the normalized intercept parameter of the Gamma model of raindrop size distribution, whereas $D_m$ is the mass-weighted mean diameter (Bringi and Chandrasekar, 2001). We have clarified this in the revision (page 1, line 16 in the clean version): *"The mean values of the normalized intercept parameter ($log_{10}N_w$) and the mass-weighted mean diameter ($D_m$) of convective rain are higher than that of stratiform rain, and there is a clear boundary between the two types of rain in terms of the scattergram of $log_{10}N_w$ versus $D_m$."*

*2. The Introduction needs to be further strengthened. It seems that this study only differs from previous studies simply through using a long-term dataset, as can be inferred from the current version, which is actually not.*
**Response:** We thank the reviewer for this great advice. We totally agree with the reviewer that this study differs from previous studies not only on the utilization of long-term raindrop size distribution data, but also the detailed analysis. For example, the impacts of urban heat island (UHI) effect on rainfall microphysical properties have never been studied in the literature. We have clarified this in the revision. Motivated by the reviewer's comment, we have also extensively revised the introduction section of this manuscript.

*3. I would suggest not to mention "local microphysics" in P2, Line 4, as apparently this present study does not provide much interpretation of rainfall microphysics. The main objective is for better characterizations of DSD in urban region and potential improvement for radar QPE.*
**Response:** We thank the reviewer for the comment. We have rewritten the introduction as suggested, although we would like to note that the characteristics of DSD are among the most important microphysical properties of local precipitation.

*4. P1, Line 21, what does "UHI up stage of a day" mean? Please clarify.*
**Response:** We thank the reviewer for pointing this out. Basically, "UHI up stage of a day" means a stage characterized by an abrupt rise of urban heat island intensity of a day (Yang et al., 2013). We have clarified this in the revision (page 1, line 21-22, in the clean version), now this sentences read: *In addition, at the stage characterized by an abrupt rise of urban heat island (UHI) intensity as well as the stage of strong UHI intensity during the day, DSD shows*

*higher $D_m$ values and lower $log_{10}N_w$ values."*

*5. Since there is a dual-pol radar collocated with the disdrometer, I wonder how the dual-pol radar fields are utilized in this study. The dual-pol fields used in this study are simulated using the T-matrix method. How accurate is the simulation?*

**Response:** We thank the reviewer for this great comment. Unfortunately, the dual-pol radar has not been deployed during this study period. There is another dual-pol radar nearby, which is managed by Beijing Meteorological Bureau (BMB). But that radar is still suffering from signal processing and data quality issue. In this study, we meant to use the simulated dual-polarized radar fields to derive the rainfall estimators, in support of the future operational X-band radar applications. The simulation is based on real raindrop size distribution data collected by the disdrometer. In particular, the scattering properties of raindrops are computed using T-matrix method (Leinonen, 2014). The accuracy of computation is 1e-3. In fact, the simulated fields as such are often used to calibrate and validate real radar (remote sensing) measurements since they are considered *in situ* measurements.

*6. Hail contamination remains a challenge for radar QPE. However, this is how dual-pol radar can surpass conventional radar (using the KDP field). It seems strange to me that the authors remove hail from all their records, as this will degrade the significance of their study. Please justify.*

**Response:** We thank the reviewer for this very good question. There are two main issues in radar quantitative precipitation estimation. One is the derivation of theoretical or experimental radar rainfall relations, and the other is real application of the derived relations. In general, only the liquid rain should be included in the algorithm development (since the ultimate goal is to conduct rainfall estimation). That is why the hail contaminated data are eliminated in the theoretical analysis.

In real applications, in order to get the liquid rainfall estimates especially from the rain-hail mixture (i.e., with hail contaminations), the *R-KDP* relations are suggested since they are not sensitive to hail compared to reflectivity *Z.* In such cases, reflectivity values, as a power term, are often very large (higher than 55 dBZ) due to hail contamination, which will lead to an overestimation of rain. On the contrary, *KDP*, as a phase term, is directly related to the liquid water content, and we can get more accurate rainfall rates using the *R-KDP* relationship. However, the choice of *R-KDP* in real applications does not mean we would need to include the hail contamination data in the derivation of theoretical algorithms. In addition, we would like to focus on the liquid rainfall properties in this study. Hail and/or winter precipitation such as snow will be investigated in future studies. We have clarified this in the revision.

*7. The threshold of 5 mm/h for separating convective and stratiform rainfall is small compared to previous studies. Please justify.*

**Response:** We thank the reviewer for pointing this out. To separate convective and stratiform rainfall, we use a combination of two thresholds, i.e., rain rate and the standard deviation of rain rate. This method has been widely used in previous studies. In particular, a threshold of 1.5mm/h on the standard deviation of rain rate is often used, and a threshold of 1.5 mm/h (Wen et al., 2019;Wen et al., 2016) or 5 mm/h (Bringi et al., 2003;Chen et al., 2013;Seela et al., 2017;Seela et al., 2018;Tang et al., 2014;Wen et al., 2017) or 10 mm/h (Marzano et al., 2010;Testud et al., 2001;Thurai et al., 2010) on rain rate is often used. In most studies in China,

the threshold of 5 mm/h is applied (Chen et al., 2013;Seela et al., 2017;Tang et al., 2014;Wen et al., 2017). In addition, the early and end stages of convective rain may be excluded from the dataset if a threshold of 10 mm/h is adopted, since the rain rates at the beginning or near ending of a convective storm are likely less than 10 mm/h (Chen et al., 2013). Based on this, we decide to use the threshold of 5 mm/h in the separation analysis.

*8. Please remove the texts P7, Lines 9-12. They can be moved to the caption of Figure 5. Similarly for P9, Lines 4-6.*
**Response:** We thank the reviewer for this suggestion. We totally agree with the reviewer. Changed as suggested!

*9. Figure 5, caption, what does "shallow events" mean? Please explain*
**Response:** We thank the reviewer for pointing this out. Shallow precipitation is a third type of precipitation besides convective and stratiform suggested by a few researchers, based on data from vertically pointing radar observations. "Shallow events" are typically characterized by low cloud top (below 0 °C isotherm) and weak rainfall rate (Fabry and Zawadzki, 1995;Cha et al., 2009). We have clarified this in the revision (page 26, line 9-10, in the clean version).
In the study by Wen et al. (2016), they used the vertical profile of reflectivity from Micro Rain Radar (MRR) and DSDs from the 2DVD to identify the shallow events. In that study, the top of radar echo of shallow rain is too low to reach the melting layer, which means that the precipitation forms directly in liquid form and no melting is present (Fabry and Zawadzki, 1995;Cha et al., 2009). The corresponding DSDs of this shallow rain have a relatively small maximum diameter and high concentration of raindrops with small diameters, indicating distinctions among the microphysical processes of the three precipitation types. In our study, due to the lack of vertical measurements, we focus on the convective and stratiform precipitation.

*10. Figure 5 and texts, I'm not sure if it is reasonable to compare this study with previous studies, as clearly this study present climatological features of DSD, while the referenced studies seem to be event-based.*
**Response:** We thank the reviewer for raising this concern. Although previous studies seem event-based, they essentially represent the local climatology and microphysics of different precipitation types. Therefore, we believe it is useful to conduct such comparison. In addition, this study provides new evidence from Asia (northern China) to further support the DSD analysis in the mid-latitudes.

*11. I would suggest to present frequency distribution of rain rates among different UHI stages, along with DSD parameters in Figure 9. As the authors explained differences of DSD parameters for different rain rates in previous section, differences of DSD parameters among UHI stages might be simply due to rain rate differences. This suggestion also applies for the analysis of seasonal cycle in section 3.5.*
**Response:** We thank the reviewer for this great suggestion. We have revised the manuscript as suggested. In particular, the frequency distribution of rain rates for different UHI stages and different months is supplemented. Descriptions of these two parts have been rephrased as follows: "*The DSD spectra of different diurnal periods are quite similar to those of different rain rate classes, showing a unimodal shape and peak position at the diameter $D \sim 0.5$ mm.*

*It is notable that the DSD spectra are almost the same at small drop size bins (D < 1 mm) and have the same width. As the diameter becomes larger, variations in the DSD spectra start showing up. The DSD spectra of S UHI stage and UHI U stage show similar and higher concentration, whereas the DSD spectra of W UHI stage and UHI D stage have similar but lower concentration, indicating that during the UHI U stage and S UHI stage, high-intensity rainfall is more likely to occur. This is in line with the study in Yang et al. (2017), which showed that the short term high-intensity rainfall was more likely to happen at the UHI U stage and end at the late S UHI stage. The frequency and variation of rain rate for different UHI stage (see Fig. S2) can also indicate this point."*

*"As shown in Fig. 11, all the DSD spectra have a peak at diameter D ~ 0.5 mm, which are consistent with other classifications in this study. The DSD in May has a relatively higher concentration while a relatively lower concentration in July. At small drop size bins (D < 1 mm), the spectra for May and September are similar, while the spectra for other four months are similar. As the diameter increases, the differences between these spectra become larger, and the DSD spectrum for July has the highest concentration and October the lowest concentration. The rainfall with higher concentration and large drops is more likely to happen in July, leading to a high rain rate intensity (see also Fig. S3). "*

[Figure]

Figure S2: Histograms of rain rate $\log_{10} R$ ($R$ in mm h$^{-1}$) at different UHI stages: (a) UHI down stage; (b) weak UHI stage; (c) UHI up stage; (d) strong UHI stage; (e) variation of rain rate $R$ (mm h$^{-1}$) for different UHI stages. The white central lines in the boxes indicate the medians. The black central lines indicate the means, and the bottom and top lines of the box indicate the 25th and 75th percentiles, respectively. The bottom and top lines of the vertical lines out of the box indicate the 5th and 95th percentiles, respectively.

[Figure]

Figure S3: Same as Figure S2, but for different months.

*12. Grammar and wording need double check. There are some typos throughout the manuscript, for instance, "P1,Line 34, warn should be warm", etc*

**Response:** We appreciate the reviewer's careful reading of this manuscript. We have double checked the Grammar and wording issues in this manuscript. We have also asked a colleague (a native English speaker) to perform an additional internal review of this manuscript.

**References:**

Bringi, V.N. and Chandrasekar, V., 2001. Polarimetric Doppler weather radar: principles and applications. Cambridge university press, PP410.

Bringi, V. N., Chandrasekar, V., Hubbert, J., Gorgucci, E., Randeu, W. L., and Schoenhuber, M.: Raindrop size distribution in different climatic regimes from disdrometer and dual-polarized radar analysis, J. Atmos. Sci., 60, 354-365, Doi 10.1175/1520-0469(2003)060<0354:Rsdidc>2.0.Co;2, 2003.

Cha, J.-W., Chang, K.-H., Yum, S. S., and Choi, Y.-J. J. A. i. A. S.: Comparison of the bright band characteristics measured by Micro Rain Radar (MRR) at a mountain and a coastal site in South Korea, 26, 211-221, 2009.

Chen, B. J., Yang, J., and Pu, J. P.: Statistical Characteristics of Raindrop Size Distribution in the Meiyu Season Observed in Eastern China, J. Meteorolog. Soc. Jpn., 91, 215-227, 10.2151/jmsj.2013-208, 2013.

Fabry, F., and Zawadzki, I.: Long-term radar observations of the melting layer of precipitation

and their interpretation, J. Atmos. Sci., 52, 838-851, 1995.

Leinonen, J.: High-level interface to T-matrix scattering calculations: architecture, capabilities and limitations, Opt. Express, 22,1655–1660, doi:10.1364/OE.22.001655, 2014.

Marzano, F. S., Cimini, D., and Montopoli, M. J. A. R.: Investigating precipitation microphysics using ground-based microwave remote sensors and disdrometer data, 97, 583-600, 2010.

Seela, B. K., Janapati, J., Lin, P. L., Reddy, K. K., Shirooka, R., and Wang, P. K.: A Comparison Study of Summer Season Raindrop Size Distribution Between Palau and Taiwan, Two Islands in Western Pacific, J. Geophys. Res-Atmos., 122, 11787-11805, 10.1002/2017jd026816, 2017.

Seela, B. K., Janapati, J., Lin, P. L., Wang, P. K., and Lee, M. T.: Raindrop Size Distribution Characteristics of Summer and Winter Season Rainfall Over North Taiwan, J. Geophys. Res-Atmos., 123, 11602-11624, 10.1029/2018jd028307, 2018.

Tang, Q., Xiao, H., Guo, C. W., and Feng, L.: Characteristics of the raindrop size distributions and their retrieved polarimetric radar parameters in northern and southern China, Atmos. Res., 135, 59-75, 10.1016/j.atmosres.2013.08.003, 2014.

Testud, J., Oury, S., Black, R. A., Amayenc, P., and Dou, X. J. J. o. A. M.: The concept of "normalized" distribution to describe raindrop spectra: A tool for cloud physics and cloud remote sensing, 40, 1118-1140, 2001.

Thurai, M., Bringi, V., May, P. J. J. o. A., and Technology, O.: CPOL radar-derived drop size distribution statistics of stratiform and convective rain for two regimes in Darwin, Australia, 27, 932-942, 2010.

Wen, G., Xiao, H., Yang, H. L., Bi, Y. H., and Xu, W. J.: Characteristics of summer and winter precipitation over northern China, Atmos. Res., 197, 390-406, 10.1016/j.atmosres.2017.07.023, 2017.

Wen, L., Zhao, K., Zhang, G. F., Xue, M., Zhou, B. W., Liu, S., and Chen, X. C.: Statistical characteristics of raindrop size distributions observed in East China during the Asian summer monsoon season using 2-D video disdrometer and Micro Rain Radar data, J. Geophys. Res-Atmos., 121, 2265-2282, 10.1002/2015jd024160, 2016.

Wen, L., Zhao, K., Wang, M., and Zhang, G. J. A. i. A. S.: Seasonal Variations of Observed Raindrop Size Distribution in East China, 36, 346-362, 2019.

Yang, P., Ren, G. Y., and Liu, W. D.: Spatial and Temporal Characteristics of Beijing Urban Heat Island Intensity, J. Appl. Meteorol. Climatol., 52, 1803-1816, 10.1175/Jamc-D-12-0125.1, 2013.

---

## Author Comment (AC2) · 21 Aug 2019

We thank the reviewer for the kind words. We appreciate all the valuable comments and suggestions provided by the reviewer. We have carefully revised this manuscript based on the reviewer's comments. Detailed responses are attached in a separate document.

Please also note the supplement to this comment: https://www.hydrol-earth-syst-sci-discuss.net/hess-2019-210/hess-2019-210-AC2-supplement.pdf

210, 2019.

**Supplement:**

**Response to Reviewer 2**

*General Comments:*

*In summary, this study analyzed the statistical characteristics of raindrop size distribution (DSD) during rainy seasons (May-October) in Beijing based on a 5-year observation (2014-2018) from a Parsivel2 disdrometer deployed at Tsinghua University, compared the differences in diameter and concentration between rain types, rainfall intensity, urban heat island (UHI) stages and months, and finally explored its implications for two types of radar rainfall estimations. The manuscript is overall detailed and well written with analysis of DSD parameters and suggestions for precipitation forecast, while it has some minor problems and lacks further explanation of precipitation micro physics. Therefore, I suggest a minor revision and encourage the authors to improve this manuscript. Detailed suggestions are listed below. As I'm not working on this specific researching area, some suggestions may not be suitable for this manuscript, and the authors can decide whether or not to accept them.*

**Response:** We thank the reviewer for the kind words. We appreciate all the valuable comments and suggestions provided by the reviewer. We have carefully revised this manuscript based on the reviewer's comments. In the text below we quote the reviewers' comments verbatim and we follow them with our detailed responses in red.

*Major Comments:*

*1. I've noticed the authors actually show their results together with discussions in Section 3 and 4, while I personally prefer an independent Discussion Section to clarify the differences and significance of this study compared to others on DSD characteristics in Beijing (and other cities). For example, the authors derived an opposed conclusion referred to Wen and Zhang's work (P7 L10), and it would be better if the authors mark their observation locations in Fig. 1(b), explain the differences in physical mechanism and show detailed possible causes.*

**Response:** We thank the reviewer for this great suggestion. We have mark the observation locations in Wen et al., (2017) and Ji et al., (2019) in Beijing in fig. 1. The study by Tang et al. (2014) did not detail their disdrometer position clearly, just with a description of position: "Beijing".

[Figure]

Figure 1: (a) the topography of Beijing, (b) the locations of DSD studies in Beijing area, the red mark represents the location of Parsivel[2] disdrometer deployed at Tsinghua University

campus in this study, the green and purple makers represent locations in the studies by Wen et al., (2017) and Ji et al., (2019), respectively.

The comparison of DSDs in different part of China (i.e., North China, East China, and South China) are indicated in Fig. 5. Even in the same region, the DSDs measured by different instruments have notable differences, such as the differences in Beijing between results from Wen et al. (2017) (2DVD, circle) and Tang et al. (2014) (Parsivel, square). In order to reduce the errors caused by different measurement instruments, in our study, only DSDs measured by Parsivel disdrometers are analyzed. It is concluded that the east part of China has the lowest mean value of $\log_{10}N_w$ (3.42) with highest mean value of $D_m$ (1.66), while southern China has the highest mean value of $\log_{10}N_w$ (3.86) with middle value of $D_m$ (1.46), and the north part of China has the middle value of $\log_{10}N_w$ (3.60) with lowest value of $D_m$ (1.15). There are also differences between Beijing in this study and studies in other parts of China (Wen et al. (2016) in eastern China and Zhang et al. (2019) in southern China). These differences indicates that the DSD characteristics are highly correlated to the specific geographical locations and associated climate regimes.

For Beijing area, the results of this study and Tang et al. (2014) show great differences in convective rain and less differences in stratiform rain. These may be attributed to different convective systems during different years, and the limited measurements from only one season in the study by Tang et al. (2014), which are not sufficient to represent local DSD characteristic. However, we want to note that the detailed comparison in microphysical mechanisms of rainfall is not the main focus of this study, although results from previous studies are briefly summarized. As mentioned, this study presents more of climatological features of local DSD in Beijing, while the referenced studies seem to be event-based. More data would be required to resolve the detailed differences in physical mechanism, which can be a good future study.

*2. Abstract Section. I suggest the authors should first clarify the meanings before using symbols or abbreviations such as Dm and lgNw when showing results in Abstract Section. In addition, although P4 L15 defined Nw as "normalized intercept parameter", I have not found its clear physical meaning which expected to be similar to Nt, the total number concentration.*

**Response:** We thank the reviewer for this very important comment. $D_m$ is the mass-weighted mean diameter and $\log_{10}N_w$ is the normalized intercept parameter of a Gamma model of raindrop size distribution (Bringi and Chandrasekar, 2001). We have clarified this in the Abstract Section (P1 line 16 in the clean version). In addition, $N_t$ $(\mathrm{m}^{-3})$ is the total number concentration, representing an integral of the rain drop size distribution at all diameters, and it is different for the distribution parameter $N_w$ $(\mathrm{m}^{-3}\ \mathrm{mm}^{-1})$. The relationship of these two parameter is:

$$N_t = \int N(D)dD = \int N_w f(\mu) \left(\frac{D}{D_m}\right)^{\mu} \exp\left[-(4+\mu)\frac{D}{D_m}\right] dD$$

We have clarified this in the revised manuscript (Page 5, line 4 in the clean version).

*3. P6, L15, the authors use specific mean and standard derivation values of rain rate (R) as thresholds to separate convective rain from stratiform rain. However, it seems that R is only related to D spectra considering equation (10) and (3), so in my opinion this classification method is equivalent to solving nonlinear equations and will probably cause the "clear*

*boundary" in DSD characteristics between rain types mentioned in Abstract Section. The authors should pay attention to the classification method chosen in this study, and it would be better if they obtain more information on rain types from other data sources.*

**Response:** We totally agree with the reviewer that the classification method may cause a "clear boundary" in the DSD characteristics since both $R$, $\log_{10}N_w$ and $D_m$ are derived from the raindrop size spectra. We can get the relationship among these three parameters with a power law velocity assumption by Atlas and Ulbrich (1977),

$$v(D) = 3.78D^{0.67}; m/s$$

$$R = (0.6 \times 10^{-3}\pi)(3.78)N_w f(\mu)\Gamma(\mu + 4.67)\frac{D_m^{4.67}}{(4 + \mu)^{\mu+4.67}}; mm/h$$

As such, other data sources such as reflectivity profiles are used to classify the rain type in several studies (Cha et al., 2009;Wen et al., 2016). However, it was found that there was no significant differences compared to using $R$ only, and using other data sources may cause different issues since they are not directly related to rainfall intensity (rain rate estimation algorithm should be applied). In addition, since $\log_{10}N_w$ and $D_m$ are different moments of the raindrop spectra compared to the rainfall rate. The "clear boundary" is not really as sharp as one would expect. Provided the ground disdrometer data, the thresholds of mean and standard derivation values are still the most commonly used way to classify rainfall type (Bringi et al., 2003;Chen et al., 2013). Motivated by the reviewer's comment, we have revised the manuscript by highlighting the potential of using auxiliary data in the classification of different rainfall types (last paragraph Page13 line 30-32, in the clean version): *We also want to note that combining additional observations such as the vertically-pointing profiler radar data (White et al., 2003) can further enhance the classification results of different rainfall types, which should be considered in future studies."*

*4. There is a mistake in Table 5. The correct UHI stage labels in the table should be UHI D, W UHI, UHI U and S UHI, which is consistent with Figure 9 and 10 indicating UHI W stage has the largest mean concentration and lowest Dm.*

**Response:** We apologize for this mistake. We have corrected this in the revision. The corrected version is listed below for the reviewer's information. Thanks again for pointing this out.

Table 5: Mean and Standard Deviation (SD) Values of $R$, $D_m$, $\log_{10}N_w$, $N_t$, $W$, $\mu$, and $\Lambda$ for different diurnal periods based on UHI intensity

| | $R$(mm h$^{-1}$) | | $D_m$ (mm) | | $\log_{10}N_w$ (m$^{-3}$ mm$^{-1}$) | | $N_t$ (m$^{-3}$) | | $W$ (g m$^{-3}$) | | $\mu$ | | $\Lambda$ | |
|---|---|---|---|---|---|---|---|---|---|---|---|---|---|---|
| | Mean | SD | Mean | SD | Mean | SD | Mean | SD | Mean | SD | Mean | SD | Mean | SD |
| UHI D | 1.88 | 4.31 | 1.11 | 0.42 | 3.59 | 0.60 | 342.15 | 499.30 | 0.10 | 0.19 | 15.06 | 13.63 | 9.32 | 8.49 |
| W UHI | 2.04 | 4.10 | 1.10 | 0.41 | 3.70 | 0.58 | 378.44 | 398.08 | 0.12 | 0.18 | 15.27 | 14.48 | 9.33 | 8.90 |
| UHI U | 2.82 | 6.94 | 1.18 | 0.51 | 3.57 | 0.65 | 380.88 | 488.27 | 0.15 | 0.30 | 14.09 | 13.45 | 8.78 | 8.45 |
| S UHI | 2.60 | 6.79 | 1.18 | 0.46 | 3.56 | 0.64 | 385.00 | 563.30 | 0.14 | 0.30 | 13.97 | 13.95 | 8.61 | 8.43 |

*5. Figure 13. This figure may mislead the readers as the study focused mainly on low rain rate values (less than 25 mm/h). I suggest the authors should plot it on double logarithmic coordinates, which will make it a linear relationship (i.e. convert Z=238R^1.57 to lgZ=1.57lgR+lg238). Besides, the derived line for total rainfall are below both convective and stratiform lines for low rain rate values, and the authors should explain this.*

**Response:** We thank the reviewer for this very good suggestion. We agree with the reviewer that the double logarithmic plot for *Z-R* relationship might be better. We have revised the figure

and rephrased the related descriptions in the main manuscript (From Page 11 line 12-19 in the clean version). The revised figure is repeated here for the reviewer information.

[Figure]

**Figure 13: Scatter density plot of $R$ (mm h$^{-1}$) versus $Z_H$ (mm$^6$ m$^{-3}$) for all rain events. The black, red, and blue curves respectively stand for the fitted power-law relations for total rain, convective rain, and stratiform rain. The purple and green dashed lines denote the default NEXRAD $Z - R$ relation (Fulton et al., 1998) and a commonly used continental stratiform rain relation (Marshall and Palmer, 1948), respectively.**

*6. Section 4.1 and 4.2. How did the authors figure out the relationship equations (14)-(17)? In my opinion, it is more likely that the uncertainty in parameter values, other than suitability of algorithms, may be the main sources of normalized absolute error (NMAE).*

**Response:** We thank the reviewer for this great comment. The relationships in equations (14)-(17) are derived through nonlinear regression using the least square method. We have clarified this in the revision (page 11, line 29 in the clean version). In the nonlinear fitting processing, we attempted to minimize the uncertainty induced by the parameter values. Such power-law relations are typically used by weather radars for quantitative precipitation estimation. Therefore, the uncertainty in the parameter values are essentially the same with the "suitability" of radar rainfall algorithms (or maybe the reviewer is referring to something else?). This type of uncertainty is also called "parameterization" error (Bringi and Chandrasekar, 2001). The values of *NMAE* can be an indicator of such parameterization error of different algorithms. We have clarified this in the revision (From page 12 line 5-10, in the clean version).

*Minor Comments:*
*7. P2, L19-26. These sentences are weird to read with duplicate words such as "high spatial and temporal variabilities". I guess the authors here wanted to elaborate the complexity of*

*measuring and modeling precipitation in Beijing due to its high urbanization (i.e. densely populated) and large heterogeneity (i.e. high spatial and temporal variabilities), and show the significance of analyzing DSD characteristics which could help us to understand urban precipitation. I suggest that the authors should rewrite this part to keep it concise and clear.*

**Response:** We apologize for the possible confusion. We have rephrased these sentences as suggested. Now it reads: *"The rapid urbanization and complex topography have further exacerbated the high variability of precipitation in Beijing urban area, posing challenges to precipitation observations and forecast (Song et al., 2014; Yang et al., 2013a; Yang et al., 2016). This also highlights the importance of understanding local DSD characteristics to better quantify the urban precipitation."* (page 2, lines 25-28, in the clean version)

*8. P2, L21, ": : : stations network de Vos et al., 2017", add "by" after "network". In addition, I prefer a standard usage of references in the text.*

**Response:** We thank the reviewer for this comment. In the revision, we have added a "by" after "network". In addition, we have standardized the references and formatting in the text.

*9. P2, L22, "monitoring networks : : : have been applied", here using "established" may be a better choice.*

**Response**: We totally agree with the reviewer. Changed as suggested!

*10. P2, L34, "warn" -> "warm".*

**Response:** Corrected as suggested!

*11. P3, L5, "methodologies" -> "methods".*

**Response:** Changed as suggested!

*12. P3, L7. I suggest the word "Section" should be capitalized.*

**Response:** Changed as suggested!

*13. P3, L15-17 and L25. From the manuscript, I guess these 32 non-uniform bins are set by THUD and fixed for all rainfall events, leaving the maximum observable diameter to be 24.5 mm. However, P5 L20 mentioned that the biggest raindrops ever reported are around 8 mm. The authors should clearly point it out if the latter diameter value can only represent precipitation in Beijing.*

**Response:** We thank the reviewer for pointing this out. The 32 non-uniform bins are set by the second-generation Particle Size and Velocity (Parsivel$^2$) disdrometer (Loffler-Mang and Joss, 2000) and are fixed for all events. The disdrometer can not only observe raindrops but also other precipitation particles such as hail and snowflakes, which are typically larger than raindrops.

The biggest raindrop ever reported is around 8 mm (Beard et al., 1986;Baumgardner and Colpitt, 1995). Therefore, the maximum diameter is often limited to 8 mm, not only in Beijing, but also other regions in the world. This is commonly recognized in the precipitation community. We have clarified this in the revision (page 5, line 26-28 in the clean version): *"In addition, to focus on rainfall, all the data contaminated by hail are removed, and raindrops at a diameter of larger than 8 mm are eliminated (Bringi and Chandrasekar, 2001) since the biggest raindrops ever reported globally in the literature are around 8 mm (Baumgardner and Colpitt, 1995; Beard et al., 1986)."*

*14. P3, L24-25. How to obtain Dj if only the number of raindrops belonging to each bin was recorded? I've noticed that the maximum value of Dmax happened to be 7.5 mm in Table 1, so I guess there should exists a bin ranging from 7 mm to 8 mm, and the authors took its average as corresponding diameter.*

**Response:** We thank the reviewer for this detailed question. For the second-generation Particle Size and Velocity (Parsivel$^2$) disdrometer, the measured particles are subdivided into 32 different diameter bins. At each diameter bin, it has a specific mid-value and spread. In this study, we consider the mid-value as $D_j$. For example, the mid-value of the 24$^{th}$ bin is 7.5 mm and the bin spread is 1 mm, which means the raindrops in this category range from 7 mm to 8 mm. Then we take the mid-value of 7.5 mm as $D_{24}$ corresponding to this particular bin. We have further clarified this in the revision (page 4, line 9-11, in the clean version): *"where $D_j$ (mm) is the mid-value of jth diameter bin, $N(D_j)$ is in $m^{-3}\ mm^{-1}$; $A$ is the sampling area in $m^2$; $\Delta t$ is the sampling time interval in s; $A$ and $\Delta t$ are respectively 0.0054 $m^2$ and 60 s in this study; $\Delta D_j$ (mm) is the diameter spread for the jth diameter bin; $V_i$ (m s$^{-1}$) is the mid-value fall speed for the ith velocity class."*

*15. P5, L30. How did the authors figure out the relationship between Dm and D0?*

**Response:** We apologize for the possible confusion. The relationship between $D_m$ and $D_0$ is derived by Ulbrich (1983). For any reason, this reference was lost. We have clarified this in the revision (page 6, line 11 in the clean version): *"The relationship $\Lambda D_m + 3.67 = \Lambda D_0 + 4$ (Ulbrich, 1983) may explain for such phenomenon when $\Lambda > 0$."*

*16. P13, L9. I guess the authors missed "(MP-Strariform)" after "NEXRAD".*

**Response:** We thank the reviewer for pointing this out. In the revision, "(MP-Stratiform)" has been added after "NEXRAD".

**References**

Atlas, D., and Ulbrich, C. W. J. J. o. A. M.: Path-and area-integrated rainfall measurement by microwave attenuation in the 1–3 cm band, 16, 1322-1331, 1977.

Baumgardner, D. C., and Colpitt, A.: Monster drops and rain gushes: unusual precipitation phenomena in Florida marine cumulus, 1995, 15-20.

Beard, K. V., Johnson, D. B., and Baumgardner, D.: Aircraft Observations of Large Raindrops in Warm, Shallow, Convective Clouds, Geophys. Res. Lett., 13, 991-994, DOI 10.1029/GL013i010p00991, 1986.

Bringi, V. N., and Chandrasekar, V.: Polarimetric Doppler weather radar: principles and applications, Cambridge university press, 2001.

Bringi, V. N., Chandrasekar, V., Hubbert, J., Gorgucci, E., Randeu, W. L., and Schoenhuber, M.: Raindrop size distribution in different climatic regimes from disdrometer and dual-polarized radar analysis, J. Atmos. Sci., 60, 354-365, Doi 10.1175/1520-0469(2003)060<0354:Rsdidc>2.0.Co;2, 2003.

Cha, J.-W., Chang, K.-H., Yum, S. S., and Choi, Y.-J. J. A. i. A. S.: Comparison of the bright band characteristics measured by Micro Rain Radar (MRR) at a mountain and a coastal site in South Korea, 26, 211-221, 2009.

Chen, B. J., Yang, J., and Pu, J. P.: Statistical Characteristics of Raindrop Size Distribution in

the Meiyu Season Observed in Eastern China, J. Meteorolog. Soc. Jpn., 91, 215-227, 10.2151/jmsj.2013-208, 2013.

Ji, L., Chen, H., Li, L., Chen, B., Xiao, X., Chen, M., and Zhang, G. J. R. S.: Raindrop Size Distributions and Rain Characteristics Observed by a PARSIVEL Disdrometer in Beijing, Northern China, Remote Sens, 11, 1479, 2019.

Loffler-Mang, M., and Joss, J.: An optical disdrometer for measuring size and velocity of hydrometeors, J. Atmos. Oceanic Technol., 17, 130-139, Doi 10.1175/1520-0426(2000)017<0130:Aodfms>2.0.Co;2, 2000.

Operating instructions Present Weather Sensor OTT Parsivel2, 2016, OTT Messtechnik, Germany. Available at: https://www.ott.com/download/operating-instructions-present-weather-sensor-ott-parsivel2-without-screen-heating/

Tang, Q., Xiao, H., Guo, C. W., and Feng, L.: Characteristics of the raindrop size distributions and their retrieved polarimetric radar parameters in northern and southern China, Atmos. Res., 135, 59-75, 10.1016/j.atmosres.2013.08.003, 2014.

Ulbrich, C. W.: Natural Variations in the Analytical Form of the Raindrop Size Distribution, J. Climate Appl. Meteorol., 22, 1764-1775, Doi 10.1175/1520-0450(1983)022<1764:Nvitaf>2.0.Co;2, 1983.

Wen, G., Xiao, H., Yang, H. L., Bi, Y. H., and Xu, W. J.: Characteristics of summer and winter precipitation over northern China, Atmos. Res., 197, 390-406, 10.1016/j.atmosres.2017.07.023, 2017

Wen, L., Zhao, K., Zhang, G. F., Xue, M., Zhou, B. W., Liu, S., and Chen, X. C.: Statistical characteristics of raindrop size distributions observed in East China during the Asian summer monsoon season using 2-D video disdrometer and Micro Rain Radar data, J. Geophys. Res-Atmos., 121, 2265-2282, 10.1002/2015jd024160, 2016.

White, A.B., P.J. Neiman, F.M. Ralph, D.E. Kingsmill, and P.O. Persson, 2003: Coastal Orographic Rainfall Processes Observed by Radar during the California Land-Falling Jets Experiment. J. Hydrometeor., 4, 264–282

Zhang, A. S., Hu, J. J., Chen, S., Hu, D. M., Liang, Z. Q., Huang, C. Y., Xiao, L. S., Min, C., and Li, H. W.: Statistical Characteristics of Raindrop Size Distribution in the Monsoon Season Observed in Southern China, Remote Sens, 11, 432, 10.3390/rs11040432, 201

---

## Author Comment (AC3) · 21 Aug 2019

The revised manuscript-clean version is attached as a separate document.

Please also note the supplement to this comment: https://www.hydrol-earth-syst-sci-discuss.net/hess-2019-210/hess-2019-210-AC3-supplement.pdf
* * *

---

## Author Comment (AC4) · 21 Aug 2019

The revised manuscript with track changes is attached as a separate document.

Please also note the supplement to this comment:
https://www.hydrol-earth-syst-sci-discuss.net/hess-2019-210/hess-2019-210-AC4-supplement.pdf
* * *

---

## Author Comment (AC5) · 21 Aug 2019

**Statistical characteristics of raindrop size distribution during rainy seasons in Beijing urban area and implications for radar rainfall estimation**

Yu Ma[1], Guangheng Ni[1], Chandrasekar V.Chandra[2], Fuqiang Tian[1], Haonan Chen[2,3]

[revised manuscript text omitted]

20   Beijing, the capital of China, is a very densely populated metroplex with a population higher than 21 million. It is more vulnerable to extreme weather events such as torrential rainfall and floods (Zhang et al., 2013). Since the hydrology response in urban area is sensitive to the spatial and temporal variability of rainfall (Cristiano et al., 2017), rainfall monitoring networks with high-temporal and spatial resolution (e.g., dense network of automatic weather stations by de Vos et al. (2017); remote sensing network described by Chen and Chandrasekar (2015) and Cifelli et al. (2018)) have been established in several

25   metropolitan areas. The rapid urbanization and complex topography have further exacerbated the high variability of precipitation in Beijing urban area, posing challenges to precipitation observations and forecast (Song et al., 2014; Yang et al., 2013a; Yang et al., 2016). This also highlights the importance of understanding local DSD characteristics to better quantify the urban precipitation.

Several studies on DSD characteristics in Beijing area have been conducted. Tang et al. (2014) studied the DSD

30   characteristics and the polarimetric radar parameters for convective and stratiform rain from July to October 2008 and compared with other regions using a first-generation laser-based optical particle size and velocity (Parsivel[1]) disdrometer manufactured by OTT Messtechnik, Germany. Wen et al. (2017a) investigated the statistical properties of summer and winter precipitation in Beijing, including the bulk properties, raindrop fall velocity, axis ratio, and DSD, using a two-dimensional

video disdrometer (2DVD) and a micro-rain radar (MRR). Ji et al. (2019) analyzed the microphysical structure of DSD using 14-month DSD measurements from a second-generation Particle Size and Velocity (Parsivel[2]) disdrometer in Beijing.

However, these studies are mainly focused on summer time (June-September or July-October) or with very limited measurements from one season or two, which are not sufficient to represent local DSD characteristics, especially the monthly variability, during the rainy seasons ranging from May to October. In addition, the impacts of urban heat island (UHI) effect on rainfall microphysical properties have never been studied in the literature, as the DSD measurements used in previous studies are more likely collected in the suburban area.

This paper presents a comprehensive study of DSD properties using 5-year (2014–2018) continuous observations in Beijing urban area, aiming to advance our understanding and characterizations of DSD in urban region, as well as parameterization in remote sensing retrievals and NWP models. The DSD properties, their variabilities, as well as the potential applications in radar QPE are detailed.

[revised manuscript text omitted]

The derived X-band radar rainfall relations are as follows:

$$R(Z_H) = 0.0576Z_H^{0.557}, \tag{14}$$

$$R(K_{\mathrm{dp}}) = 15.421K_{\mathrm{dp}}^{0.817}, \tag{15}$$

$$R(K_{\mathrm{dp}}, Z_{\mathrm{DR}}) = 26.778K_{\mathrm{dp}}^{0.946}Z_{DR}^{-1.249}, \tag{16}$$

$$R(Z_H, Z_{\mathrm{DR}}) = 5.886 \times 10^{-3}Z_H^{0.994}Z_{DR}^{-4.929}, \tag{17}$$

5      Note that there are differences in the $Z - R$ relationships between X- and S-band due to Mie scattering at higher frequency. Previous studies showed that the parameterization errors associated with various radar rainfall relations are among the key factors affecting the derived rainfall performance (Bringi and Chandrasekar, 2001). Hence, the parameterization errors in the X-band radar rainfall algorithms are investigated and quantified in this study. Figure 14 illustrates the scatter density plots of rain rates derived from $R(Z_H)$, $R(K_{\mathrm{dp}})$, $R(K_{\mathrm{dp}}, Z_{\mathrm{DR}})$, and $R(Z_H, Z_{\mathrm{DR}})$ versus the rain rates directly computed from DSD. To

10   quantify the parameterization errors, the normalized mean absolute error (NMAE) of estimated rainfall rate is calculated, which is defined as:

$$\mathrm{NMAE} = \frac{\langle|R_{EP}-R_D|\rangle}{\langle R_D\rangle}, \tag{18}$$

where the angle brackets stand for sample average; $R_{EP}$ and $R_D$ denote the estimated rain rates derived from parameterized radar rainfall algorithms and DSD information, respectively. The $NMAE_{RR}$ is calculated for different rainfall rate intervals

15   from 0 to 100 mm h$^{-1}$. Figure 15 shows the parameterization error structure of $R(Z_H)$, $R(K_{\mathrm{dp}})$, $R(K_{\mathrm{dp}}, Z_{\mathrm{DR}})$, and $R(Z_H, Z_{\mathrm{DR}})$ as a function of rainfall rate.

      It can be seen from Figs. 14 and 15 that the algorithms based on dual polarization radar parameters can provide better estimates than $Z - R$ relationship. In addition, the dual parameter algorithms, namely $R(K_{\mathrm{dp}}, Z_{\mathrm{DR}})$ and $R(Z_H, Z_{\mathrm{DR}})$, have even better performance than the single parameter based algorithm including $R(K_{\mathrm{dp}})$. The NMAE has a decreasing trend as the rain

20   rate increase from 1 mm h$^{-1}$ to 60 mm h$^{-1}$. The fluctuation when rain rate is greater than 60 mm h$^{-1}$ may be due to the random errors caused by few samples of large values. The parameterization errors of $R(K_{\mathrm{dp}})$, $R(K_{\mathrm{dp}}, Z_{\mathrm{DR}})$, and $R(Z_H, Z_{\mathrm{DR}})$ become stable when rain rate is getting higher than 10 mm h$^{-1}$. It is also noted that at low rain rate (less than 10 mm h$^{-1}$), the NMAE of $R(Z_H, Z_{\mathrm{DR}})$ is the smallest, while at higher rain rate (higher than 10 mm h$^{-1}$) the NMAE of $R(K_{dp}, Z_{\mathrm{DR}})$ becomes the smallest. This again highlights the importance of selecting appropriate rain rate relations for local radar applications.

25   **5 Summary and Conclusion**

In this paper, 5-year (2014–2018) observations of DSD from a disdrometer deployed at Tsinghua University are analyzed to explore the microphysical characteristics of precipitation during rainy seasons (May–October) in Beijing urban area. The main conclusions are as follows:

[revised manuscript text omitted]

---

## Author Comment (AC6) · 21 Aug 2019

**Statistical characteristics of raindrop size distribution during rainy seasons in Beijing urban area and implications for radar rainfall estimation**

Yu Ma[1], Guangheng Ni[1], Chandrasekar V.Chandra[2], Fuqiang Tian[1], Haonan Chen[2,3]

[revised manuscript text omitted]

  Beijing, the capital of China, is a very densely populated metroplex with a population higher than 21 million. It is more vulnerable to extreme weather events such as torrentialextreme rainfall and floods (Zhang et al., 2013). Since the hydrology response in urban area is sensitive to the spatial and temporal variability of rainfall (Cristiano et al., 2017), Rainfall rainfall monitoring networks with high-temporal and spatial resolution (e.g., dense network of automatic weather stations network by de Vos et al., (2017); remote sensing network described by Chen and Chandrasekar (2015) and Cifelli et al., (2018)) have been applied established in several metropolitan areas., as the hydrology response in urban area is sensitive to the spatial and temporal variability of rainfall (Cristiano et al., 2017). The precipitation in Beijing is more complex with high spatial and temporal variability due to the combined effects of high-urbanization and local unique topography (Song et al., 2014; Yang et al., 2013a; Yang et al., 2016), which highlights the importance of further understanding of DSD characteristics for enhanced urban precipitation measurements and modelling. The rapid urbanization and complex topography have further exacerbated the high variability of precipitation in Beijing urban area, posing challenges to precipitation observations and forecast (Song et al., 2014; Yang et al., 2013a; Yang et al., 2016). This also highlights the importance of understanding local DSD characteristics to better quantify the urban precipitation.

Several studies on DSD characteristics in Beijing area have been conducted. Tang et al. (2014) studied the DSD characteristics and the polarimetric radar parameters for convective and stratiform rain from July to October 2008  and compared with other regions using a first-generation laser-based optical particle size and velocity (Parsivel[1]) disdrometer  manufactured by OTT Messtechnik, Germany. Wen et al. (2017a) investigated the statistical properties of summer and winter precipitation in Beijing, including the bulk properties, raindrop fall velocity, axis ratio, and DSD, using a two-dimensional video disdrometer (2DVD) and a micro-rain radar (MRR). Ji et al. (2019) analyzed the microphysical structure of DSD using 14-month DSD measurements from a second-generation Particle Size and Velocity (Parsivel[2]) disdrometer in Beijing.

However, these studies are mainly focused on summer time (June-September or July-October)  or with very limited measurements from one season or two, which are not sufficient to represent local DSD characteristics, especially the monthly variability,  during the rainy  seasons ranging from May to October. In addition, the impacts of urban heat island (UHI) effect on rainfall microphysical properties have never been studied in the literature, as the DSD measurements used in previous studies are more likely collected in the suburban area.

This paper presents a comprehensive study of DSD properties using 5-year (2014–2018) continuous observations in Beijing urban area, aiming to advance our understanding and characterizations of DSD in urban region, as well as parameterization in remote sensing retrievals and NWP models. The DSD properties, their variabilities, as well as the potential applications in radar QPE are detailed.

[revised manuscript text omitted]

small, which also suggests that the DSD may be under size-controlled conditions (Steiner et al., 2004). Meanwhile, the relationship for total rainfall ($Z = 238R^{1.57}$) underestimates the rain rate at low values compared with the stratiform relationship ($Z = 171R^{2.15}$), due to the inconsistent rain rate - reflectivity structures of two rain types.

The default NEXRAD algorithm and MP-Stratiform relationship for continental stratiform rain are also indicated in Fig.
13 for comparison.  At low reflectivity values ($Z_H <$ 23 dBZ), the curve of MP-Stratiform relationship is below  the local stratiform relation, but at higher values, it reverses. As the mean reflectivity of stratiform rain (21 dBZ) is less than 23 dBZ (See Table 2), the MP-Stratiform relationship may introduce underestimation of rainfall. The default NEXRAD relationship behaves similarly: underestimation at lower reflectivity values and overestimation at higher reflectivity values. Considering the mean reflectivity value of convective rain, the default NEXRAD relationship may cause overestimation of rainfall. In other words, the default relationship $Z = 300R^{1.4}$ should be used with caution for local applications in Beijing.

**4.2 High frequency (X-band) polarimetric radar applications**

A high-resolution dual-polarization X-band radar network is being deployed for urban hydrometeorological applications in Beijing area. To support the radar deployment and facilitate the rainfall applications, the polarimetric parameters, including differential reflectivity $Z_{dr}$ (dB) and specific differential propagation phase shift $K_{dp}$($°\mathrm{km}^{-1}$) are computed from the DSD measurements. Therein, the $T$-matrix method (Waterman, 1965) is adopted and the computations are made  at X-band frequency.  In addition, the polarimetric rainfall relations are derived based on the nonlinear least-squares method, including $R(K_{\mathrm{dp}})$, $R(K_{\mathrm{dp}}, Z_{\mathrm{DR}})$, and $R(Z_H, Z_{\mathrm{DR}})$. Here $Z_{\mathrm{DR}} = 10^{Z_{dr}/10}$ is the differential reflectivity in linear scale.

$$R(Z_H) = 0.0304Z_H^{0.638}, \tag{14}$$

$$R(K_{\mathrm{dp}}) = 15.421K_{\mathrm{dp}}^{0.817}, \tag{15}$$

$$R(K_{\mathrm{dp}}, Z_{DR}) = 26.778K_{\mathrm{dp}}^{0.946}Z_{DR}^{-1.249}, \tag{16}$$

$$R(Z_H, Z_{DR}) = 4.785 \times 10^{-3}Z_H^{0.978}Z_{DR}^{-3.226}, \tag{17}$$

The derived X-band radar rainfall relations are as follows:

$$R(Z_H) = 0.0576Z_H^{0.557}, \tag{14}$$

$$R(K_{\mathrm{dp}}) = 15.421K_{\mathrm{dp}}^{0.817}, \tag{15}$$

$$R(K_{\mathrm{dp}}, Z_{\mathrm{DR}}) = 26.778K_{\mathrm{dp}}^{0.946}Z_{DR}^{-1.249}, \tag{16}$$

$$R(Z_H, Z_{\mathrm{DR}}) = 5.886 \times 10^{-3}Z_H^{0.994}Z_{DR}^{-4.929}, \tag{17}$$

Note that there are differences in the $Z - R$ relationships between X- and S-band due to Mie scattering at higher frequency.

[revised manuscript text omitted]

---

## Author Comment (AC7) · 21 Aug 2019

Revised Supplementary Material is attached as a separate document.

Please also note the supplement to this comment:
https://www.hydrol-earth-syst-sci-discuss.net/hess-2019-210/hess-2019-210-AC7-supplement.pdf
* * *

---

## Author Comment (AC8) · 21 Aug 2019

**Supplement**

[Figure]

**Figure S1: Scatter density plot of $\log_{10}N_w$ versus $D_0$: (a) the total rainfall events; (c) stratiform events; (d) convective events. (b) is the scatterplot of $\log_{10}N_w$ versus $D_0$ for convective (red circle dots) and stratiform (blue square dots) cases. The black dashed line is the $\log_{10}N_w - D_0$ relationship for stratiform rain reported by Thurai et al. (2016).**

5

[Figure]

Figure S2: Histograms of rain rate $\log_{10}R$ ($R$ in mm h$^{-1}$) at different UHI stages: (a) UHI down stage; (b) weak UHI stage; (c) UHI up stage; (d) strong UHI stage; (e) variation of rain rate $R$ (mm h$^{-1}$) for different UHI stages. The white central lines in the boxes indicate the medians. The black central lines indicate the means, and the bottom and top lines

[Figure]

of the box indicate the 25th and 75th percentiles, respectively. The bottom and top lines of the vertical lines out of the box indicate the 5th and 95th percentiles, respectively.

Figure S3: Same as Figure S2, but for different months.

5

---

## Author Response (AR1)

**Response to Reviewer #1**

*Overall comments:*

*The authors present a well-designed study of DSD over a dense urban area. The results can advance our understanding of rainfall microphysics and improve radar QPE in urban areas. There are some places in the manuscript that need further clarification, but other than that, this is a well-written paper and can be accepted after revision. My specific comments are listed below (not necessarily in order of importance).*

**Response:** We thank the reviewer for the kind words. We appreciate the reviewer's time and effort spent on our manuscript. We have carefully revised this manuscript based on the reviewer's comments. In the text below we quote the reviewers' comments verbatim and we follow them with our detailed responses in red.

*Specific comments:*

*1. Please explain the meaning of $log_{10}N_w$ and $D_m$ on their first occurrence.*

**Response:** We thank the reviewer for this suggestion. $log_{10}N_w$ is the normalized intercept parameter of the Gamma model of raindrop size distribution, whereas $D_m$ is the mass-weighted mean diameter (Bringi and Chandrasekar, 2001). We have clarified this in the revision (page 1, line 16 in the clean version): *"The mean values of the normalized intercept parameter ($log_{10}N_w$) and the mass-weighted mean diameter ($D_m$) of convective rain are higher than that of stratiform rain, and there is a clear boundary between the two types of rain in terms of the scattergram of $log_{10}N_w$ versus $D_m$."*

*2. The Introduction needs to be further strengthened. It seems that this study only differs from previous studies simply through using a long-term dataset, as can be inferred from the current version, which is actually not.*

**Response:** We thank the reviewer for this great advice. We totally agree with the reviewer that this study differs from previous studies not only on the utilization of long-term raindrop size distribution data, but also the detailed analysis. For example, the impacts of urban heat island (UHI) effect on rainfall microphysical properties have never been studied in the literature. We have clarified this in the revision. Motivated by the reviewer's comment, we have also extensively revised the introduction section of this manuscript.

*3. I would suggest not to mention "local microphysics" in P2, Line 4, as apparently this present study does not provide much interpretation of rainfall microphysics. The main objective is for better characterizations of DSD in urban region and potential improvement for radar QPE.*

**Response:** We thank the reviewer for the comment. We have rewritten the introduction as suggested, although we would like to note that the characteristics of DSD are among the most important microphysical properties of local precipitation.

*4. P1, Line 21, what does "UHI up stage of a day" mean? Please clarify.*

**Response:** We thank the reviewer for pointing this out. Basically, "UHI up stage of a day" means a stage characterized by an abrupt rise of urban heat island intensity of a day (Yang et al., 2013). We have clarified this in the revision (page 1, line 21-22, in the clean version), now this sentences read: *In addition, at the stage characterized by an abrupt rise of urban heat island (UHI) intensity as well as the stage of strong UHI intensity during the day, DSD shows*

*higher* $D_m$ *values and lower* $log_{10}N_w$ *values.''*

*5. Since there is a dual-pol radar collocated with the disdrometer, I wonder how the dual-pol radar fields are utilized in this study. The dual-pol fields used in this study are simulated using the T-matrix method. How accurate is the simulation?*

**Response:** We thank the reviewer for this great comment. Unfortunately, the dual-pol radar has not been deployed during this study period. There is another dual-pol radar nearby, which is managed by Beijing Meteorological Bureau (BMB). But that radar is still suffering from signal processing and data quality issue. In this study, we meant to use the simulated dual-polarized radar fields to derive the rainfall estimators, in support of the future operational X-band radar applications. The simulation is based on real raindrop size distribution data collected by the disdrometer. In particular, the scattering properties of raindrops are computed using T-matrix method (Leinonen, 2014). The accuracy of computation is 1e-3. In fact, the simulated fields as such are often used to calibrate and validate real radar (remote sensing) measurements since they are considered *in situ* measurements.

*6. Hail contamination remains a challenge for radar QPE. However, this is how dual-pol radar can surpass conventional radar (using the KDP field). It seems strange to me that the authors remove hail from all their records, as this will degrade the significance of their study. Please justify.*

**Response:** We thank the reviewer for this very good question. There are two main issues in radar quantitative precipitation estimation. One is the derivation of theoretical or experimental radar rainfall relations, and the other is real application of the derived relations. In general, only the liquid rain should be included in the algorithm development (since the ultimate goal is to conduct rainfall estimation). That is why the hail contaminated data are eliminated in the theoretical analysis.

In real applications, in order to get the liquid rainfall estimates especially from the rain-hail mixture (i.e., with hail contaminations), the *R-KDP* relations are suggested since they are not sensitive to hail compared to reflectivity *Z*. In such cases, reflectivity values, as a power term, are often very large (higher than 55 dBZ) due to hail contamination, which will lead to an overestimation of rain. On the contrary, *KDP*, as a phase term, is directly related to the liquid water content, and we can get more accurate rainfall rates using the *R-KDP* relationship. However, the choice of *R-KDP* in real applications does not mean we would need to include the hail contamination data in the derivation of theoretical algorithms. In addition, we would like to focus on the liquid rainfall properties in this study. Hail and/or winter precipitation such as snow will be investigated in future studies. We have clarified this in the revision.

*7. The threshold of 5 mm/h for separating convective and stratiform rainfall is small compared to previous studies. Please justify.*

**Response:** We thank the reviewer for pointing this out. To separate convective and stratiform rainfall, we use a combination of two thresholds, i.e., rain rate and the standard deviation of rain rate. This method has been widely used in previous studies. In particular, a threshold of 1.5mm/h on the standard deviation of rain rate is often used, and a threshold of 1.5 mm/h (Wen et al., 2019;Wen et al., 2016) or 5 mm/h (Bringi et al., 2003;Chen et al., 2013;Seela et al., 2017;Seela et al., 2018;Tang et al., 2014;Wen et al., 2017) or 10 mm/h (Marzano et al., 2010;Testud et al., 2001;Thurai et al., 2010) on rain rate is often used. In most studies in China,

the threshold of 5 mm/h is applied (Chen et al., 2013;Seela et al., 2017;Tang et al., 2014;Wen et al., 2017). In addition, the early and end stages of convective rain may be excluded from the dataset if a threshold of 10 mm/h is adopted, since the rain rates at the beginning or near ending of a convective storm are likely less than 10 mm/h (Chen et al., 2013). Based on this, we decide to use the threshold of 5 mm/h in the separation analysis.

*8. Please remove the texts P7, Lines 9-12. They can be moved to the caption of Figure 5. Similarly for P9, Lines 4-6.*
**Response:** We thank the reviewer for this suggestion. We totally agree with the reviewer. Changed as suggested!

*9. Figure 5, caption, what does "shallow events" mean? Please explain*
**Response:** We thank the reviewer for pointing this out. Shallow precipitation is a third type of precipitation besides convective and stratiform suggested by a few researchers, based on data from vertically pointing radar observations. "Shallow events" are typically characterized by low cloud top (below 0 °C isotherm) and weak rainfall rate (Fabry and Zawadzki, 1995;Cha et al., 2009). We have clarified this in the revision (page 26, line 9-10, in the clean version).
In the study by Wen et al. (2016), they used the vertical profile of reflectivity from Micro Rain Radar (MRR) and DSDs from the 2DVD to identify the shallow events. In that study, the top of radar echo of shallow rain is too low to reach the melting layer, which means that the precipitation forms directly in liquid form and no melting is present (Fabry and Zawadzki, 1995;Cha et al., 2009). The corresponding DSDs of this shallow rain have a relatively small maximum diameter and high concentration of raindrops with small diameters, indicating distinctions among the microphysical processes of the three precipitation types. In our study, due to the lack of vertical measurements, we focus on the convective and stratiform precipitation.

*10. Figure 5 and texts, I'm not sure if it is reasonable to compare this study with previous studies, as clearly this study present climatological features of DSD, while the referenced studies seem to be event-based.*
**Response:** We thank the reviewer for raising this concern. Although previous studies seem event-based, they essentially represent the local climatology and microphysics of different precipitation types. Therefore, we believe it is useful to conduct such comparison. In addition, this study provides new evidence from Asia (northern China) to further support the DSD analysis in the mid-latitudes.

*11. I would suggest to present frequency distribution of rain rates among different UHI stages, along with DSD parameters in Figure 9. As the authors explained differences of DSD parameters for different rain rates in previous section, differences of DSD parameters among UHI stages might be simply due to rain rate differences. This suggestion also applies for the analysis of seasonal cycle in section 3.5.*
**Response:** We thank the reviewer for this great suggestion. We have revised the manuscript as suggested. In particular, the frequency distribution of rain rates for different UHI stages and different months is supplemented. Descriptions of these two parts have been rephrased as follows: "*The DSD spectra of different diurnal periods are quite similar to those of different rain rate classes, showing a unimodal shape and peak position at the diameter  D  ~ 0.5 mm.*

*It is notable that the DSD spectra are almost the same at small drop size bins ($D < 1$ mm) and have the same width. As the diameter becomes larger, variations in the DSD spectra start showing up. The DSD spectra of S UHI stage and UHI U stage show similar and higher concentration, whereas the DSD spectra of W UHI stage and UHI D stage have similar but lower concentration, indicating that during the UHI U stage and S UHI stage, high-intensity rainfall is more likely to occur. This is in line with the study in Yang et al. (2017), which showed that the short term high-intensity rainfall was more likely to happen at the UHI U stage and end at the late S UHI stage. The frequency and variation of rain rate for different UHI stage (see Fig. S2) can also indicate this point."*

*"As shown in Fig. 11, all the DSD spectra have a peak at diameter $D \sim 0.5$ mm, which are consistent with other classifications in this study. The DSD in May has a relatively higher concentration while a relatively lower concentration in July. At small drop size bins ($D < 1$ mm), the spectra for May and September are similar, while the spectra for other four months are similar. As the diameter increases, the differences between these spectra become larger, and the DSD spectrum for July has the highest concentration and October the lowest concentration. The rainfall with higher concentration and large drops is more likely to happen in July, leading to a high rain rate intensity (see also Fig. S3). "*

[Figure]

Figure S2: Histograms of rain rate $\log_{10}R$ ($R$ in mm h$^{-1}$) at different UHI stages: (a) UHI down stage; (b) weak UHI stage; (c) UHI up stage; (d) strong UHI stage; (e) variation of rain rate $R$ (mm h$^{-1}$) for different UHI stages. The white central lines in the boxes indicate the medians. The black central lines indicate the means, and the bottom and top lines of the box indicate the 25th and 75th percentiles, respectively. The bottom and top lines of the vertical lines out of the box indicate the 5th and 95th percentiles, respectively.

[Figure]

Figure S3: Same as Figure S2, but for different months.

*12. Grammar and wording need double check. There are some typos throughout the manuscript, for instance, "P1,Line 34, warn should be warm", etc*

**Response:** We appreciate the reviewer's careful reading of this manuscript. We have double checked the Grammar and wording issues in this manuscript. We have also asked a colleague (a native English speaker) to perform an additional internal review of this manuscript.

[Figure]

Figure 1: (a) the topography of Beijing, (b) the locations of DSD studies in Beijing area, the red mark represents the location of Parsivel[2] disdrometer deployed at Tsinghua University

campus in this study, the green and purple makers represent locations in the studies by Wen et al., (2017) and Ji et al., (2019), respectively.

The comparison of DSDs in different part of China (i.e., North China, East China, and South China) are indicated in Fig. 5. Even in the same region, the DSDs measured by different instruments have notable differences, such as the differences in Beijing between results from Wen et al. (2017) (2DVD, circle) and Tang et al. (2014) (Parsivel, square). In order to reduce the errors caused by different measurement instruments, in our study, only DSDs measured by Parsivel disdrometers are analyzed. It is concluded that the east part of China has the lowest mean value of $\log_{10}N_w$ (3.42) with highest mean value of $D_m$ (1.66), while southern China has the highest mean value of $\log_{10}N_w$ (3.86) with middle value of $D_m$ (1.46), and the north part of China has the middle value of $\log_{10}N_w$ (3.60) with lowest value of $D_m$ (1.15). There are also differences between Beijing in this study and studies in other parts of China (Wen et al. (2016) in eastern China and Zhang et al. (2019) in southern China). These differences indicates that the DSD characteristics are highly correlated to the specific geographical locations and associated climate regimes.

For Beijing area, the results of this study and Tang et al. (2014) show great differences in convective rain and less differences in stratiform rain. These may be attributed to different convective systems during different years, and the limited measurements from only one season in the study by Tang et al. (2014), which are not sufficient to represent local DSD characteristic. However, we want to note that the detailed comparison in microphysical mechanisms of rainfall is not the main focus of this study, although results from previous studies are briefly summarized. As mentioned, this study presents more of climatological features of local DSD in Beijing, while the referenced studies seem to be event-based. More data would be required to resolve the detailed differences in physical mechanism, which can be a good future study.

*2. Abstract Section. I suggest the authors should first clarify the meanings before using symbols or abbreviations such as Dm and lgNw when showing results in Abstract Section. In addition, although P4 L15 defined Nw as "normalized intercept parameter", I have not found its clear physical meaning which expected to be similar to Nt, the total number concentration.*

**Response:** We thank the reviewer for this very important comment. $D_m$ is the mass-weighted mean diameter and $\log_{10}N_w$ is the normalized intercept parameter of a Gamma model of raindrop size distribution (Bringi and Chandrasekar, 2001). We have clarified this in the Abstract Section (P1 line 16 in the clean version). In addition, $N_t$ (m$^{-3}$) is the total number concentration, representing an integral of the rain drop size distribution at all diameters, and it is different for the distribution parameter $N_w$ (m$^{-3}$ mm$^{-1}$). The relationship of these two parameter is:

$$N_t = \int N(D)dD = \int N_w f(\mu) \left(\frac{D}{D_m}\right)^{\mu} \exp\left[-(4+\mu)\frac{D}{D_m}\right] dD$$

We have clarified this in the revised manuscript (Page 5, line 4 in the clean version).

*3. P6, L15, the authors use specific mean and standard derivation values of rain rate (R) as thresholds to separate convective rain from stratiform rain. However, it seems that R is only related to D spectra considering equation (10) and (3), so in my opinion this classification method is equivalent to solving nonlinear equations and will probably cause the "clear*

*boundary" in DSD characteristics between rain types mentioned in Abstract Section. The authors should pay attention to the classification method chosen in this study, and it would be better if they obtain more information on rain types from other data sources.*

**Response:** We totally agree with the reviewer that the classification method may cause a "clear boundary" in the DSD characteristics since both $R$, $\log_{10}N_w$ and $D_m$ are derived from the raindrop size spectra. We can get the relationship among these three parameters with a power law velocity assumption by Atlas and Ulbrich (1977),

$$v(D) = 3.78D^{0.67}; m/s$$

$$R = (0.6 \times 10^{-3}\pi)(3.78)N_w f(\mu)\Gamma(\mu + 4.67)\frac{D_m^{4.67}}{(4 + \mu)^{\mu+4.67}}; mm/h$$

As such, other data sources such as reflectivity profiles are used to classify the rain type in several studies (Cha et al., 2009;Wen et al., 2016). However, it was found that there was no significant differences compared to using $R$ only, and using other data sources may cause different issues since they are not directly related to rainfall intensity (rain rate estimation algorithm should be applied). In addition, since $\log_{10}N_w$ and $D_m$ are different moments of the raindrop spectra compared to the rainfall rate. The "clear boundary" is not really as sharp as one would expect. Provided the ground disdrometer data, the thresholds of mean and standard derivation values are still the most commonly used way to classify rainfall type (Bringi et al., 2003;Chen et al., 2013). Motivated by the reviewer's comment, we have revised the manuscript by highlighting the potential of using auxiliary data in the classification of different rainfall types (last paragraph Page13 line 30-32, in the clean version): *We also want to note that combining additional observations such as the vertically-pointing profiler radar data (White et al., 2003) can further enhance the classification results of different rainfall types, which should be considered in future studies."*

*4. There is a mistake in Table 5. The correct UHI stage labels in the table should be UHI D, W UHI, UHI U and S UHI, which is consistent with Figure 9 and 10 indicating UHI W stage has the largest mean concentration and lowest Dm.*

**Response:** We apologize for this mistake. We have corrected this in the revision. The corrected version is listed below for the reviewer's information. Thanks again for pointing this out.

Table 5: Mean and Standard Deviation (SD) Values of $R$, $D_m$, $\log_{10}N_w$, $N_t$, $W$, $\mu$, and $\Lambda$ for different diurnal periods based on UHI intensity

| | $R$(mm h$^{-1}$) | | $D_m$ (mm) | | $\log_{10}N_w$ (m$^{-3}$ mm$^{-1}$) | | $N_t$ (m$^{-3}$) | | $W$ (g m$^{-3}$) | | $\mu$ | | $\Lambda$ | |
|---|---|---|---|---|---|---|---|---|---|---|---|---|---|---|
| | Mean | SD | Mean | SD | Mean | SD | Mean | SD | Mean | SD | Mean | SD | Mean | SD |
| UHI D | 1.88 | 4.31 | 1.11 | 0.42 | 3.59 | 0.60 | 342.15 | 499.30 | 0.10 | 0.19 | 15.06 | 13.63 | 9.32 | 8.49 |
| W UHI | 2.04 | 4.10 | 1.10 | 0.41 | 3.70 | 0.58 | 378.44 | 398.08 | 0.12 | 0.18 | 15.27 | 14.48 | 9.33 | 8.90 |
| UHI U | 2.82 | 6.94 | 1.18 | 0.51 | 3.57 | 0.65 | 380.88 | 488.27 | 0.15 | 0.30 | 14.09 | 13.45 | 8.78 | 8.45 |
| S UHI | 2.60 | 6.79 | 1.18 | 0.46 | 3.56 | 0.64 | 385.00 | 563.30 | 0.14 | 0.30 | 13.97 | 13.95 | 8.61 | 8.43 |

*5. Figure 13. This figure may mislead the readers as the study focused mainly on low rain rate values (less than 25 mm/h). I suggest the authors should plot it on double logarithmic coordinates, which will make it a linear relationship (i.e. convert Z=238R^1.57 to lgZ=1.57lgR+lg238). Besides, the derived line for total rainfall are below both convective and stratiform lines for low rain rate values, and the authors should explain this.*

**Response:** We thank the reviewer for this very good suggestion. We agree with the reviewer that the double logarithmic plot for *Z-R* relationship might be better. We have revised the figure

and rephrased the related descriptions in the main manuscript (From Page 11 line 12-19 in the clean version). The revised figure is repeated here for the reviewer information.

[Figure]

**Figure 13: Scatter density plot of $R$ (mm h$^{-1}$) versus $Z_H$ (mm$^6$ m$^{-3}$) for all rain events. The black, red, and blue curves respectively stand for the fitted power-law relations for total rain, convective rain, and stratiform rain. The purple and green dashed lines denote the default NEXRAD $Z - R$ relation (Fulton et al., 1998) and a commonly used continental stratiform rain relation (Marshall and Palmer, 1948), respectively.**

*6. Section 4.1 and 4.2. How did the authors figure out the relationship equations (14)-(17)? In my opinion, it is more likely that the uncertainty in parameter values, other than suitability of algorithms, may be the main sources of normalized absolute error (NMAE).*

**Response:** We thank the reviewer for this great comment. The relationships in equations (14)-(17) are derived through nonlinear regression using the least square method. We have clarified this in the revision (page 11, line 29 in the clean version). In the nonlinear fitting processing, we attempted to minimize the uncertainty induced by the parameter values. Such power-law relations are typically used by weather radars for quantitative precipitation estimation. Therefore, the uncertainty in the parameter values are essentially the same with the "suitability" of radar rainfall algorithms (or maybe the reviewer is referring to something else?). This type of uncertainty is also called "parameterization" error (Bringi and Chandrasekar, 2001). The values of *NMAE* can be an indicator of such parameterization error of different algorithms. We have clarified this in the revision (From page 12 line 5-10, in the clean version).

***Minor Comments:***
*7. P2, L19-26. These sentences are weird to read with duplicate words such as "high spatial and temporal variabilities". I guess the authors here wanted to elaborate the complexity of*

*measuring and modeling precipitation in Beijing due to its high urbanization (i.e. densely populated) and large heterogeneity (i.e. high spatial and temporal variabilities), and show the significance of analyzing DSD characteristics which could help us to understand urban precipitation. I suggest that the authors should rewrite this part to keep it concise and clear.*

**Response:** We apologize for the possible confusion. We have rephrased these sentences as suggested. Now it reads: *"The rapid urbanization and complex topography have further exacerbated the high variability of precipitation in Beijing urban area, posing challenges to precipitation observations and forecast (Song et al., 2014; Yang et al., 2013a; Yang et al., 2016). This also highlights the importance of understanding local DSD characteristics to better quantify the urban precipitation."* (page 2, lines 25-28, in the clean version)

*8. P2, L21, ": : : stations network de Vos et al., 2017", add "by" after "network". In addition, I prefer a standard usage of references in the text.*

**Response:** We thank the reviewer for this comment. In the revision, we have added a "by" after "network". In addition, we have standardized the references and formatting in the text.

*9. P2, L22, "monitoring networks : : : have been applied", here using "established" may be a better choice.*

**Response**: We totally agree with the reviewer. Changed as suggested!

*10. P2, L34, "warn" -> "warm".*

**Response:** Corrected as suggested!

*11. P3, L5, "methodologies" -> "methods".*

**Response:** Changed as suggested!

*12. P3, L7. I suggest the word "Section" should be capitalized.*

**Response:** Changed as suggested!

*13. P3, L15-17 and L25. From the manuscript, I guess these 32 non-uniform bins are set by THUD and fixed for all rainfall events, leaving the maximum observable diameter to be 24.5 mm. However, P5 L20 mentioned that the biggest raindrops ever reported are around 8 mm. The authors should clearly point it out if the latter diameter value can only represent precipitation in Beijing.*

**Response:** We thank the reviewer for pointing this out. The 32 non-uniform bins are set by the second-generation Particle Size and Velocity (Parsivel$^2$) disdrometer (Loffler-Mang and Joss, 2000) and are fixed for all events. The disdrometer can not only observe raindrops but also other precipitation particles such as hail and snowflakes, which are typically larger than raindrops.

The biggest raindrop ever reported is around 8 mm (Beard et al., 1986;Baumgardner and Colpitt, 1995). Therefore, the maximum diameter is often limited to 8 mm, not only in Beijing, but also other regions in the world. This is commonly recognized in the precipitation community. We have clarified this in the revision (page 5, line 26-28 in the clean version): *"In addition, to focus on rainfall, all the data contaminated by hail are removed, and raindrops at a diameter of larger than 8 mm are eliminated (Bringi and Chandrasekar, 2001) since the biggest raindrops ever reported globally in the literature are around 8 mm (Baumgardner and Colpitt, 1995; Beard et al., 1986)."*

*14. P3, L24-25. How to obtain Dj if only the number of raindrops belonging to each bin was recorded? I've noticed that the maximum value of Dmax happened to be 7.5 mm in Table 1, so I guess there should exists a bin ranging from 7 mm to 8 mm, and the authors took its average as corresponding diameter.*

**Response:** We thank the reviewer for this detailed question. For the second-generation Particle Size and Velocity (Parsivel$^2$) disdrometer, the measured particles are subdivided into 32 different diameter bins. At each diameter bin, it has a specific mid-value and spread. In this study, we consider the mid-value as $D_j$. For example, the mid-value of the 24$^{th}$ bin is 7.5 mm and the bin spread is 1 mm, which means the raindrops in this category range from 7 mm to 8 mm. Then we take the mid-value of 7.5 mm as $D_{24}$ corresponding to this particular bin. We have further clarified this in the revision (page 4, line 9-11, in the clean version): *"where $D_j$ (mm) is the mid-value of jth diameter bin, $N(D_j)$ is in m$^{-3}$ mm$^{-1}$; A is the sampling area in m$^2$; $\Delta t$ is the sampling time interval in s; A and $\Delta t$ are respectively 0.0054 m$^2$ and 60 s in this study; $\Delta D_j$ (mm) is the diameter spread for the jth diameter bin; $V_i$ (m s$^{-1}$) is the mid-value fall speed for the ith velocity class."*

*15. P5, L30. How did the authors figure out the relationship between Dm and D0?*

**Response:** We apologize for the possible confusion. The relationship between $D_m$ and $D_0$ is derived by Ulbrich (1983). For any reason, this reference was lost. We have clarified this in the revision (page 6, line 11 in the clean version): *"The relationship $\Lambda D_m + 3.67 = \Lambda D_0 + 4$ (Ulbrich, 1983) may explain for such phenomenon when $\Lambda > 0$."*

*16. P13, L9. I guess the authors missed "(MP-Strariform)" after "NEXRAD".*

**Response:** We thank the reviewer for pointing this out. In the revision, "(MP-Stratiform)" has been added after "NEXRAD".

The $R(D)$ distributions for different diurnal periods in Fig. 9b show little difference between UHI U stage and S UHI stage, and the distributions at these two stages are lower and broader than the other two stages.  At the W UHI stage, the

25 $R(D)$ distribution is the highest and the peak is at diameter around $D \sim 0.9$ mm, and the UHI D stage almost has the same peak around $D \sim 0.9$–1 mm, while the peaks  during other two stages are at the diameter around $D \sim 1$ mm. That is, the drop size at the W UHI stage which contributes most to the accumulated rainwater is smaller than those at UHI U stage or S UHI stage. The box-whisker plots of variation of  and $\log_{10}N_w$ for each diurnal periods show the same results (see Fig. 10). The W UHI stage has the highest mean concentration and the lowest

30 mean $D_m$ value, while the UHI U stage has the largest mean $D_m$ value and the S UHI stage has the lowest mean concentration.

**3.5 DSD characteristics in different months**

To  obtain a better understanding of the seasonal variations of DSD characteristics in Beijing urban area, rain data collected in different months are analyzed. The rain rate and DSD characteristics for different months are shown in Table 6. Figure 11 illustrates the corresponding DSD spectra and $R(D)$ distributions.

5     As shown in Fig. 11, all the DSD spectra have a peak at diameter $D \sim 0.5$ mm, which are consistent with other classifications in this study. The DSD in May has a relatively higher concentration while a relatively lower concentration in July. At small drop size bins ($D < 1$ mm), the spectra for May and September are similar, while the spectra for other four months are similar. As the diameter increases, the differences between these spectra become larger, and the DSD spectrum for July has the highest concentration and October  the lowest concentration. The

10 rainfall with higher concentration and large drops is more likely to happen in July, leading to a high rain rate intensity (see also Fig. S3).

    It is also noted that the $R(D)$ distribution for each month is different from each other. The distributions of May, October, and September have a peak at diameter around $D \sim 0.9$ mm, while the distributions of June and August have a peak at diameter around $D \sim 1$ mm. The $R(D)$ distribution of July has two peaks at diameter around $D \sim 1$ mm and $D \sim 1.5$ mm. In addition,

15 the $R(D)$ distribution of July is the widest and lowest, suggesting that a wide range of moderate drops contribute mostly to the rain in July. The $D_m$ and $\log_{10}N_w$ in Fig. 12 show an interesting annual circle : the $D_m$ ($\log_{10}N_w$) first goes up and (down) then goes down (up), while  in July $D_m$ ($\log_{10}N_w$) reaches the highest (lowest) value.

**4 Implications for Radar Rainfall Estimation**

**4.1 Single polarized radar applications**

20 The power-law relationship between radar reflectivity (in mm$^6$m$^{-3}$) and rain rate (in mmh$^{-1}$) ($Z = aR^b$) is the most widely used algorithm for single polarized radar QPE (including the current operational radars in Beijing). However, the coefficient $a$ and exponent $b$ greatly rely on the DSD variability  (Bringi et al., 2003; Rosenfeld and Ulbrich, 2003; Uijlenhoet, 2001). The default $Z - R$ relationship applied for the operational Weather Surveillance Radar — 1988 Doppler (WSR-88D) systems in the United States is $Z = 300R^{1.4}$ (Fulton et

25 al., 1998), whereas $Z = 200R^{1.6}$ is commonly used in the continental area for stratiform rain (Marshall and Palmer, 1948, hereafter referred to as MP-Stratiform relationship). The more appropriate and localized $a$ and $b$ are expected to improve regional radar rainfall estimation. In the following, the localized $Z - R$ relationships for different rain types are derived by the nonlinear least square method, aiming to provide references for operational S-band radar rainfall applications in Beijing.

    Figure 13 shows the scatter density plot of rain rate versus horizontal reflectivity, as well as the fitted power-law relations

30 for different rain types.  Figure 13 shows that most of the samples are at low values where both $Z_H$ and $R$ are

small, which also suggests that the DSD may be under size-controlled conditions (Steiner et al., 2004). Meanwhile, the relationship for total rainfall ($Z = 238R^{1.57}$) underestimates the rain rate at low values compared with the stratiform relationship ($Z = 171R^{2.15}$), due to the inconsistent rain rate - reflectivity structures of two rain types.

The default NEXRAD algorithm and MP-Stratiform relationship for continental stratiform rain are also indicated in Fig.

5    13 for comparison.  At low reflectivity values ($Z_H < $ 23 dBZ), the curve of MP-Stratiform relationship is below  the local stratiform relation, but at higher values, it reverses. As the mean reflectivity of stratiform rain (21 dBZ) is less than 23 dBZ (See Table 2), the MP-Stratiform relationship may introduce underestimation of rainfall. The default NEXRAD relationship behaves similarly: underestimation at lower reflectivity values

10   and overestimation at higher reflectivity values. Considering the mean reflectivity value of convective rain, the default NEXRAD relationship may cause overestimation of rainfall. In other words, the default relationship $Z = 300R^{1.4}$ should be used with caution for local applications in Beijing.

**4.2 High frequency (X-band) polarimetric radar applications**

A high-resolution dual-polarization X-band radar network is being deployed for urban hydrometeorological applications in

15   Beijing area. To support the radar deployment and facilitate the rainfall applications, the polarimetric parameters, including differential reflectivity $Z_{dr}$ (dB) and specific differential propagation phase shift $K_{dp}$ ($°\mathrm{km}^{-1}$) are computed from the DSD measurements. Therein, the $T$-matrix method (Waterman, 1965) is adopted and the computations are made  at X-band frequency.  In addition, the polarimetric rainfall relations are derived based on the nonlinear least-squares method, including $R(K_{dp})$, $R(K_{dp}, Z_{DR})$, and $R(Z_H, Z_{DR})$. Here $Z_{DR} = 10^{Z_{dr}/10}$ is the differential reflectivity in linear scale.

20

$$\sim\sim R(Z_H) = 0.0304Z_H^{0.638}, \quad\quad\quad\quad\quad\quad\quad\quad\quad\quad (14)$$

$$\sim\sim R(K_{dp}) = 15.421K_{dp}^{0.817}, \quad\quad\quad\quad\quad\quad\quad\quad\quad\quad (15)$$

$$\sim\sim R(K_{dp}, Z_{DR}) = 26.778K_{dp}^{0.946}Z_{DR}^{-1.249}, \quad\quad\quad\quad\quad\quad (16)$$

$$\sim\sim R(Z_H, Z_{DR}) = 4.785 \times 10^{-3}Z_H^{0.978}Z_{DR}^{-3.226}, \quad\quad\quad\quad (17)$$

25   The derived X-band radar rainfall relations are as follows:

$$R(Z_H) = 0.0576Z_H^{0.557}, \quad\quad\quad\quad\quad\quad\quad\quad\quad\quad (14)$$

$$R(K_{dp}) = 15.421K_{dp}^{0.817}, \quad\quad\quad\quad\quad\quad\quad\quad\quad\quad (15)$$

$$R(K_{dp}, Z_{DR}) = 26.778K_{dp}^{0.946}Z_{DR}^{-1.249}, \quad\quad\quad\quad\quad\quad (16)$$

$$R(Z_H, Z_{DR}) = 5.886 \times 10^{-3}Z_H^{0.994}Z_{DR}^{-4.929}, \quad\quad\quad\quad (17)$$

30   Note that there are differences in the $Z - R$ relationships between X- and S-band due to Mie scattering at higher frequency. Previous studies showed that the parameterization errors associated with various radar rainfall relations are among the key factors affecting the derived rainfall performance (Bringi and Chandrasekar, 2001). Hence, the parameterization errors in

[revised manuscript text omitted]

6. The localized $Z - R$ relationship derived from local DSD in Beijing is quite different from the operational NEXRAD algorithm (MP-Stratiform) which may overestimate (underestimate) rainfall at high (low) rain intensity.that may underestimate (overestimate) rainfall at low (high) rain intensity. The error structures of different algorithms show that the polarimetric radar rainfall relations $R(K_{dp})$, $R(K_{dp}, Z_{DR})$, and $R(Z_H, Z_{DR})$ have greater potential than $Z - R$ methods for urban QPE.

The statistical analysis of DSD characteristics presented in this study not only provides a further understanding of precipitation microphysical variabilities in Beijing but also provides indications for future model development to improve local precipitation forecast. In addition, a high-resolution X-band dual polarization radar network is being deployed in Beijing. This study is expected to provide references for future development of localized radar rainfall algorithms. Nevertheless, the DSD spectra also show the limitations of Parisvel$^2$ disdrometer in measuring small raindrops. Future study should be carried out with multiple instruments including a two-dimensional video disdrometer just deployed in this area. We also want to note that combining additional observations such as the vertically-pointing profiler radar data (White et al., 2003) can further enhance the classification results of different rainfall types, which should be considered in future studies. In addition, further investigation on the spatial variability of DSD induced by the complex micro-topography in urban area should be conducted in a future study.

**Data availability.**

Disdrometer data used in this study are available through contacting the authors.

**Author contributions.**

YM, GN, FT and HC conceived the idea; GN and FT provided financial support and observation data; YM conducted the detailed analysis; HC and CVC provided comments on the analysis; all authors contributed to the writing and revisions.

**Completing interests.**

The authors declare that they have no conflict of interest.

**Acknowledgments.**

[revised manuscript text omitted]

of the box indicate the 25th and 75th percentiles, respectively. The bottom and top lines of the vertical lines out of the box indicate the 5th and 95th percentiles, respectively.

[Figure]

**Figure S3: Same as Figure S2, but for different months.**